

# Critical Assessment of Geoengineering Strategies using Response Theory

Tamás Bódai[1,2], Valerio Lucarini[1,2,3], and Frank Lunkeit[3]

[1]Centre for the Mathematics of Planet Earth, University of Reading, UK
[2]Department of Mathematics and Statistics, University of Reading, UK
[3]CEN, Meteorological Institute, University of Hamburg, Germany

**Correspondence:** T. Bódai (t.bodai@reading.ac.uk)

**Abstract.** We investigate in an intermediate-complexity climate model (I) the applicability of linear response theory to assessing a geoengineering method, and (II) the success of the considered method. The geoengineering problem is framed here as a special optimal control problem, which leads mathematically to the following inverse problem. A given rise in carbon dioxide concentration $[CO_2]$ would result in a global climate change with respect to an appropriate ensemble average of the surface air temperature $\langle [T_s] \rangle$. We are looking for a suitable modulation of solar forcing which can cancel out the said global change, or modulate it in some other desired fashion. It is rather straightforward to predict this solar forcing, considering an infinite time period, by linear response theory, and we will spell out an iterative procedure suitable for numerical implementation that applies to finite time periods too.

Regarding (I), we find that under geoengineering, i.e. the combined greenhouse and solar forcing, the actual response $\Delta \langle [T_s] \rangle$ asymptotically is not zero, indicating that the linear susceptibility is not determined correctly. This is due to a significant quadratic nonlinearity of the response under system identification achieved by a forced experiment. This nonlinear contribution can in fact be easily removed, which results in much better estimates of the linear susceptibility, and, in turn, in a five-fold reduction in $\Delta \langle [T_s] \rangle$ under geoengineering. Regarding (II), however, we diagnose this geoengineering method to result in a considerable spatial variation of the surface temperature anomaly, reaching more than 2 [K] at polar/high latitude regions upon doubling the $[CO_2]$ concentration, even in the ideal case when the geoengineering method was successful in canceling out the response in the global mean. In the same time, a new climate is realised also in terms of e.g. an up to 4 [K] cooler tropopause or drier/disrupted Tropics, relative to unforced conditions.

## 1 Introduction

Geoengineering concepts with the purpose of mitigating climate change are receiving nowadays increasing attention (Allen et al., 2014; National Research Council, a, b) (http://www.ce-conference.org/) because of the potential for an enormous gain: fixing one of the greatest societal challenges primarily of a diplomatic nature, but also because of the great risk that such an unprecedented endeavour entails. However, the body of the presently available scientific analysis, albeit increasing (Lenton and Vaughan, 2013; Ferraro et al., 2014), is yet lacking the consideration of many more crucial aspects of the problem. For example, ours is the first attempt to conceptually categorize and frame geoengineering in terms of response theory (Kubo,



1966; Ruelle, 2009) and the theory of nonautonomous dynamical systems (Sell, 1967a, b; Romeiras et al., 1990; Crauel and Flandoli, 1994; Crauel et al., 1997; Arnold, 1998; Kloeden and Rasmussen, 2011; Carvalho et al., 2013), and then in turn as an inverse problem. This can be of little surprise, as these mathematical tools, although having been introduced to climate science for decades (Leith, 1975; Bell, 1980; Nicolis et al., 1985), are far from being exhausted, still finding many applications of

tackling problems in climate science in general (Cionni et al., 2004; Gritsun and Branstator, 2007; Kirk-Davidoff, 2009; Majda et al., 2010; Cooper et al., 2013; Lucarini and Sarno, 2011; Ragone et al., 2016; Lucarini et al., 2017; Herein et al., 2015, 2017; Bódai and Tél, 2012; Drótos et al., 2015, 2016). In the following we summarise briefly the existing mathematical tools (Sec. 1.1), and then frame the geoengineering problem and motivate our specific study (Sec. 1.2).

## 1.1 Elements of response theory

In *nonautonomous dissipative dynamical systems*, like the climate system, given in the form

$$\dot{x} = F(x) + \epsilon g(x) f(t) \tag{1}$$

the *response* of the system to an external forcing $f(t)$ can be *unambiguously* defined in terms of the so-called *snapshot attractor* (Romeiras et al., 1990) of the system, and the natural probability distribution or the measure $\mu_t(dx)$ supported by. Both the attractor and the measure are *unique* objects; they are defined by an *ensemble* of trajectories initialized

in the *infinite* past. The time-dependence of the snapshot attractor, also called a pullback attractor (Crauel and Flandoli, 1994; Arnold, 1998; Chekroun et al., 2011), and its measure give what is often termed as the 'forced response' (https://www.gfdl.noaa.gov/blogheld/3-transient-vs-equilibrium-climate-responses/), and the 'geometrical details' of theirs at any instant describe (statistical aspects of) the *internal variability* in a conceptually sound sense (Drótos et al., 2015).

For a scalar observable $\Psi(x)$ too the (forced) response is uniquely given by a projection of the measure. *Ruelle's response*

*theory* (Ruelle, 2009) asserts that the most basic ensemble-based statistics, the mean $\langle\Psi\rangle = \int \mu_t(dx)\Psi(x)$, can be decomposed into linear ($j=1$) and nonlinear ($j>1$) contributions:

$$\Delta\langle\Psi\rangle(t) = \langle\Psi\rangle(t) - \langle\Psi\rangle_0 = \sum_{j=1}^{\infty} \epsilon^j \langle\Psi\rangle^{(j)}, \tag{2}$$

where the $\langle\Psi\rangle^{(j)}$'s can be expressed as multiple *convolution integrals* involving the pertinent *Green's functions* (Lucarini et al., 2017). The first-order, i.e., linear, term reads as follows:

$$\langle\Psi\rangle^{(1)}(t) = G_\Psi^{(1)}(t) * f(t) = \int_{-\infty}^{\infty} d\tau\, G_\Psi^{(1)}(\tau) f(t-\tau), \tag{3}$$

where the Green's function has been established by Ruelle to take the form of

$$G_\Psi^{(1)}(t) = \int dx\, \Psi(x)(\exp[tL_f][L_g\bar{\mu}])(x), \tag{4}$$

where $\bar{\mu}(dx)$ is the natural invariant measure/probability distribution of the autonomous system ($f=0$), and the operators are defined as $L_f\mu = -\text{div}(f\mu)$ and $L_g\mu = -\text{div}(g\mu)$, in the notation of (Abramov and Majda, 2008).



The convolution integral under (3) can be *interpreted* in a way that the forcing $f(t)$ is decomposed into an infinite sequence of impulses, whereby the responses of the different impulses – that can be superimposed – are all given by the Green's function, whose first nonzero values occur at the time of the corresponding impulses. Although a single such *finite* impulse does not produce a nonzero response, a *continuous* sequence apparently can. Or, a single impulse of infinite magnitude, formally a

Dirac delta, can also produce a response, which is clearly the Green's function itself. If the continuous train of finite impulses all have the same unit magnitude, thereby forming a step function, formally the Heaviside step function $\Theta(t)$, the response is just the integral of the Green's function. Conversely, the Green's function is the derivative of the response to a unit step function. The latter prompts a numerical way of determining the Green's function, while a Dirac delta forcing is not realisable numerically.

Taking the Fourier transform (FT) of Eq. (3) we have, via the convolution theorem (Katznelson, 1976), a response formula in frequency domain:

$$\langle\Psi\rangle^{(1)}(\omega)=\chi_\Psi^{(1)}(\omega)f(\omega), \tag{5}$$

where $\chi_\Psi^{(1)}(\omega)=\mathrm{FT}[G_\Psi^{(1)}(t)]$ is called the linear *susceptibility*. This equation looks more useful for practical purposes as it dictates a simple multiplication instead of evaluating a convolution integral. However, in Sec. 2.1 we explain why this is not

the case, which is of course to do with the transformations between time and frequency domains.

## 1.2 The geoengineering problem

It has been proposed (National Research Council, b) that the effect of greenhouse forcing can be mitigated by applying another external forcing to the Earth system, by some geoengineering means, that has, in a way, an 'opposing' effect. There are various forcing types that can achieve this, but we will consider those that can be modeled by a modulation of the solar constant. We

will call this simply the "solar forcing", but it is also called "solar-radiation management" (Ricke et al., 2010, 2012). Clearly, these are means that modulate the shortwave incoming radiation. Readily proposed geoengineering methods include: a fleet of reflective satellites of large Sun-facing surface area put into orbit around Earth, aerosols sprayed into the atmosphere, artificially generated clouds, etc. A modulated solar constant model represents these geoengineering scenarios with a various degree of approximation.

Formally, the problem involves a forced/nonautonomous system, where at least two terms contribute to the forcing. For simplicity, we consider the case of only two forcing terms, and that they are both additive:

$$\dot{x}=F(x)+\epsilon(g_g(x)f_g(t)+g_s(x)f_s(t)), \tag{6}$$

where the subscripts indicate already the physical means of the forcings; 'g' for 'greenhouse' and 's' for 'solar'. Also, it is up to us to assign a value to the "small" parameter $\epsilon$, and in order to obtain a result in the uncomplicated form of (9), we choose

the same $\epsilon$ for both forcing components. The first-order contribution $\langle\Psi_\Sigma\rangle^{(1)}(t)$ of the *total response* $\Delta\langle\Psi_\Sigma\rangle$ under combined forcing, i.e., geoengineering, can be written as the superposition of first-order contributions of respective responses to the two



forcings in two separate scenarios when these forcings are acting alone:

$$\langle\Psi_\Sigma\rangle^{(1)}(t) = G^{(1)}_{\Psi,g}(t) * f_g(t) + G^{(1)}_{\Psi,s}(t) * f_s(t), \tag{7}$$

whose FT is of course

$$\langle\Psi_\Sigma\rangle^{(1)}(\omega) = \chi_{\Psi,g}(\omega)f_g(\omega) + \chi_{\Psi,s}(\omega)f_s(\omega). \tag{8}$$

Note that the nonlinear response is more complicated with multiple forcings present than a sum of multiple convolution integrals (Lucarini et al., 2017) as in the single forcing scenario.

If the 'forward' problem is the prediction of the response under a given forcing, then the *inverse* problem of 'predicting' the necessary forcing for a desired response seems to be well-defined in view of the above equations. To a linear approximation the necessary or required forcing is:

$$f_s(\omega) \approx \frac{\Delta\langle\Psi_\Sigma\rangle(\omega) - \chi_{\Psi,g}(\omega)f_g(\omega)}{\chi_{\Psi,s}(\omega)}. \tag{9}$$

For the above $\epsilon = 1$ is taken. We continue to discuss the solution of the inverse problem in Sec. 2.3, including the situation when a finite time period is considered. That situation can be interpreted as a control problem, which is in fact a rather special type of *optimal* control. This way the required forcing can be 'predetermined' which need not be updated during its application. We note that this is the first time the so-called solar-radiation management (SRM) is formulated as the solution of an inverse

problem. In previous studies (Ricke et al., 2010, 2012) the solar forcing was constructed on the basis of some models of how much radiative forcing a sudden change of some greenhouse gas concentration or the stratospheric optical depth would yield. In addition, a scenario ensemble of SRMs was created, and a selection of the most effective SRMs was made.

This inverse/control problem would have a 'direct' practical relevance had we got $f_g(t)$ a given, as assumed. However, this is clearly not the case; predicting the greenhouse gas emissions is an extremely complicated and rather daunting task, as it

is determined among others by *social* processes, for which we do not have good models. Nevertheless, efforts are underway (https://crescendoproject.eu/research/theme-4/). The current standard practice to 'deal' with this challenge, as reflected by the IPCC reports (Allen et al., 2014), is considering half a dozen 'methodologically constructed' 21st century emission scenarios. This way, instead of climate predictions one produces so-called climate *projections* belonging to hypothetical future emission scenarios. Therefore, the solution to our inverse problem has a rather *indirect* practical relevance; we can carry out at least

*scenario analyses*. The reader can find elsewhere (MacMartin et al., 2014) the description and analysis of a *feedback* control problem of *direct* practical relevance, when the solar forcing is being determined 'on the fly' with the use of some controller, adapting to a progressing greenhouse forcing, trying to realise the desired response *approximately*. Note that under feedback control, in a scenario analysis setting, a new simulation needs to be run for each emission scenario.

We point out that in e.g. Eq. (9) we write $\Psi$ denoting a generic observable. This means that we can *choose* a particular

(scalar) observable which we desire to evolve in a particular way. With a reference to the classic term of 'global warming', in contrast with 'climate change', we will attempt to enforce the cancellation of the global average surface air temperature (Sec. 3.1). With the increasingly wide-ranging analyses of climate change scenarios, however, it is clear that 'climate change'




should have a comprehensive meaning, not just a synonym for 'global warming' (Conway, 5 December 2008). In fact, physical quantities other than temperature could have a larger social or ecological impact (Allen et al., 2014). Beside the *physical type* of the observable quantity, we can have different choices with respect to the *spatial scale* of the quantity, such as local, or regional (Sec. 3.1.3), zonal (Sec. 3.1.2), global (Sec. 3.1.1), etc. averages.

Once an observable $\Psi$ is chosen to evolve in a particular way, the evolution of any other observable $\Phi$ will be *a given* – the solution of a *forward* problem formally identical to (8):

$$\langle \Phi_\Sigma \rangle^{(1)}(t) = G^{(1)}_{\Phi,g}(t) * f_g(t) + G^{(1)}_{\Phi,s}(t) * f_s(t), \tag{10}$$

with $f_s$ given, of course, by (9). Clearly, $\langle \Phi_\Sigma \rangle^{(1)}(t) \neq \langle \Psi_\Sigma \rangle^{(1)}(t)$ when $G_{\Phi,g}(t) \neq G_{\Psi,g}(t)$ and/or $G_{\Phi,s}(t) \neq G_{\Psi,s}(t)$, which is the generic case. Regarding the desire of cancellation $\Delta\langle \Psi_\Sigma \rangle = 0$, we can frame geoengineering – considering for simplicity
only quasistatically slow changes $f_g(t)$ – as a confinement to the 0 isoline of $\Delta\langle \Psi_\Sigma \rangle$ over the plane of $f_g$ and $f_s$ (Lucarini). In general, this isoline is different for different observables $\Phi \neq \Psi$, that is, under linear response these straight isolones fan out of the origin of the $f_g$-$f_s$ plane. This is illustrated in Fig. 1, where the curvature of the isolines for larger values of $f_g$ and $f_s$ reflect also the more general situation of nonlinear responses. It is implied then that when the system is confined to one isoline, it can obviously not be confined to the different isolines of other variables $\Phi_i$; that is, (unwanted) changes $\Delta\langle \Phi_{i,\Sigma} \rangle \neq 0$ will
ensue. In other words: the proposed geoengineering method will provide just a partial solution at best. While one aspect of the problem is solved, other aspects can be neglected, or even changed to the worse, possibly with catastrophic consequences.[1] This possibility is the main *motivation* of our present very preliminary investigation following this introductory conceptual clarification. Having enforced (approximately, to various degrees) a cancellation of global average surface air temperature, $\Delta\langle \Psi_\Sigma \rangle = \Delta\langle [T_{s,\Sigma}] \rangle \approx 0$, we will *diagnose* unwanted changes (response) in terms of:

– $\Phi = [T_s]_\lambda$ – zonal (Sec. 3.1.2) and

– $\Phi = T_s$ – regional averages on the surface, and

– $\Phi = T_{tr}$ – regional averages near the troposphere/tropopause (Sec. 3.1.3), and

– $\Phi = [P_y]$ and $P_y$ – annual precipitation (Sec. 3.2).

Note that we denote spatial averaging by square brackets, subscripted by the spatial variable(s) with respect to which we
average over its whole range, e.g. longitudes $\lambda$ for zonal averages, and for areal/global averaging we drop the subscripting (instead of writing e.g. $[T_s]_{\lambda,\mu}$).

We point out that the greenhouse and solar forcings have been found approximately "equivalent" in terms of the stationary response of the global average surface air temperature (Boschi et al., 2013) insomuch that its isolines are parallel straight lines (even if there is a curvature of the surface). This was found to be the case in rather extensive ranges of the forcings, 1200-1500

---

[1]Furthermore, we note that, as it is often acknowledged, 'no-one is living under the average climate'. Although, some live closer than others. That is, while the primary problem can be solved for some, even that will not be solved for others. Therefore, the debate on climate engineering is unlikely to have less political overtone and motive than the climate debate itself.





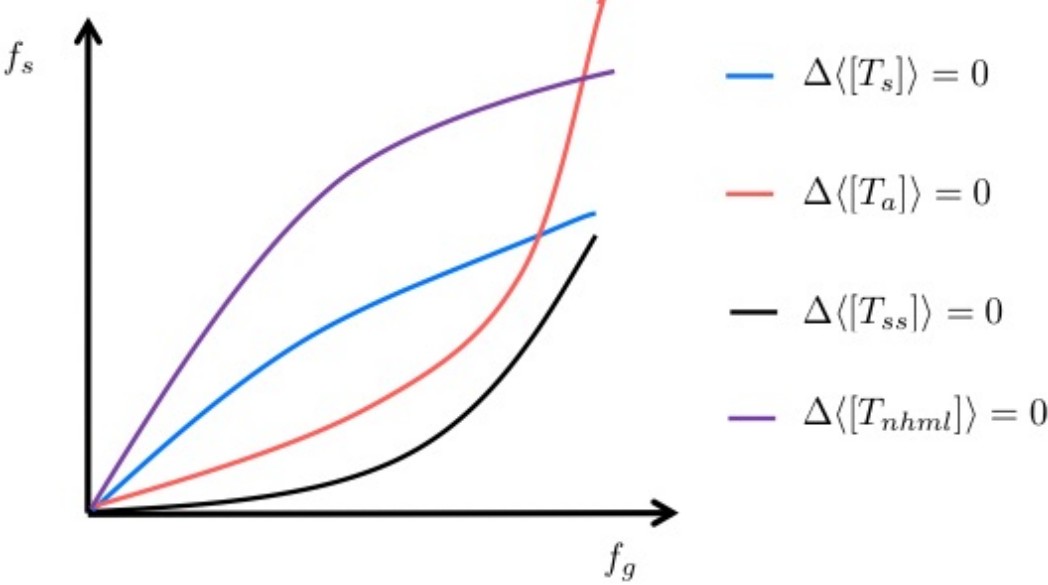

**Figure 1.** A cartoon of hypothetical isolines in the plane of greenhouse and solar forcings $f_g$-$f_s$ for various observables: $\Delta\langle[T_s]\rangle = 0$ – globally averaged surface air temperature, $\Delta\langle[T_a]\rangle = 0$ – globally averaged atmospheric temperature, $\Delta\langle[T_{ss}]\rangle = 0$ – averaged sea surface temperature, $\Delta\langle[T_{nhml}]\rangle = 0$ – surface air temperature averaged on the midlatitudes of the Northern hemisphere (reproduction of Fig. 5 of (Lucarini)).

Wm$^{-2}$ and 90-2880 ppm, respectively. That is, any curvature of the blue line as shown in Fig. 1 occurs outside of the said ranges. However, it is an open question if these forcings are equivalent in the same sense in terms of other variables too, and we will demonstrate that it is indeed not the case.

This work follows (Ragone et al., 2016) and (Lucarini et al., 2017). In the latter it has been demonstrated that response
5   theory can predict spatial patterns, which, as outlined above, is one type of diagnostics that we use to assess the success of the geoengineering method. In both of these works the demonstrations were carried out on the intermediate-complexity GCM the Planet Simulator, or PlaSim (Fraedrich, 2012), but with slightly differing setups. Here we adopt the setup of (Lucarini et al., 2017) featuring meridional ocean heat transport. The present work also builds on (Gritsun and Lucarini, 2017) adopting a simple technique to obtain a better estimate of the linear susceptibility. It finds its use especially so in our context of aiming to
10  cancel effects of greenhouse forcing by solar forcing, where the response under combined forcing is found – as also in (Boschi et al., 2013) – to be approximately linear for magnitudes of the greenhouse forcing for which, when applied separately, the response is already considerably nonlinear. This implies that in the latter situation, as considered in (Ragone et al., 2016; Lucarini et al., 2017; Gritsun and Lucarini, 2017), a more accurate susceptibility estimate would not be more productive.



We point out that the examined model PlaSim is lacking many realistic features, such as e.g. seasonal forcing or a deep ocean. The former deficiency results in very large global average surface temperature responses (Ragone et al., 2016), and the latter one does not allow for long time scales, typically of the order of hundred years. However, our technique is applicable in principle also to models with such long time scales. What is more, it would handle such situations powerfully given that any time *horizon* can be imposed on the analysis, constructing *transient* responses only, without the need of running very long experiments in which a new steady climate emerges upon external forcing. What makes this possible is that the Green's function is needed to be determined up to times only up to which we want to determine the response, as indicated by Eq. (3). We wish also to clarify that our analysis technique requires the estimation of the Green's function, which is most straightforward to do by subjecting the system to external identification forcing (Sec. 2.2), which is clearly not possible in the case of Earth. Our analysis technique is intended rather for an efficient scenario analysis in a *model*, where the side-effects of geoengineering can be calculated for any given emission scenario using negligible computer resources. When one places themselves in the ideal setting of knowing the Green's function of a realistic model, any side-effect of the considered geoengineering concept to be found, according to our objective (II), would serve as a warning that in reality the side-effect would most likely be worse.

The structure of the remainder is as follows. Next in Sec. 2 we detail our methodology: the notation and algorithm for spectral analysis in discrete time (Sec. 2.1), the way we obtain the Green's functions (Sec. 2.2), our novel solution method to the inverse problem for a required solar forcing (Sec. 2.3), and a zoo of experiments used to assess nonlinearities and else (Sec. 2.4).[2] Then in Sec. 3 we provide results: firstly, pertaining to objective (I), about the success of the primary objective of geoengineering, the cancellation (Sec. 3.1.1), and then, pertaining to objective (II), our diagnostics of other observables (Secs. 3.1.2, 3.1.3, 3.2). Finally, in Sec. 4, in terms of the stationary climate only, (I) we outline an improved method of obtaining the required solar forcing for cancellation, and also (II) provide some diagnostics for the case of exact cancellation. In Sec. 5 we summarize our results and give our perspective of worthwhile future work.

## 2 Methodology

### 2.1 Computing the response in time and frequency domains

To be able to carry out (approximate) calculations involving spectral transforms, we need to clarify the formulae and algorithms applicable to *discrete time* and *finite size* data. We can approximate the time-continuous strictly monotonically evolving forcing $f(t)$ by a *staircase-like* forcing that is defined by a uniform *sampling* of $f(t)$, called a *sample-and-hold* approximation. It can be represented by a discrete sequence $f[n] = f(t = nT), n = \ldots, -1, 0, 1, \ldots, T$ being the uniform sampling interval, in which sequence the data points provide the levels of the "steps". That is, for an actual staircase-like forcing signal $f(t = (n+\nu)T) = f[n]$ for all $\nu \in [0, 1]$, where the noninteger $\nu$ can be viewed as a phase variable – the phase where the sample is taken within the interval where the forcing is constant. For such staircase-like forcings sample values of the response with the sampling

---

[2]The reader who is not concerned with computational aspects can skip Secs. 2.1, 2.2, 2.3. However, Sec. 2.4 is unavoidable in order to understand how the results presented subsequently will enable us to make conclusions regarding (I) the applicability of linear response theory.





$\Psi[n] = \Psi(t = (n + \nu)T)$ at any phase $\nu \in [0, 1]$ obey:

$$\langle \hat{\Psi} \rangle^{(1)}[n] = \sum_{k=-\infty}^{\infty} h_\Psi[k] f[n-k] = h_\Psi[n] * f[n] \qquad (11)$$

where the discrete-time (DT) impulse response or DT Green's function $h_\Psi[n]$ is, clearly, the response $\langle \hat{\Psi}_\perp \rangle^{(1)}$ to a Kronecker delta function forcing: $f[n] = \delta[n] = 1$ if $n = 0$ and 0 otherwise (Hespanha, 2009). Note that we make a distinction in our notation with regard to the special forcing such that we distinguish $\hat{\Psi}$ from $\Psi$; however, for simplicity, we did not subscript $\hat{\Psi}$ by $\nu$ despite that it depends on the phase. Note also that in general $h_\Psi[n] \neq G_\Psi^{(1)}[n] = G_\Psi^{(1)}(t = (n + \nu)T)$ with the same $\nu$ as $\Psi[n]$ (or $\hat{\Psi}[n]$) is defined with, or with any $\nu$ and all $n$. Clearly, once the sampling frequency is not adequate regarding some 'strongly featured' time scales of the forcing, the calculated discrete response will be also an inadequate approximation. We note further that – unlike the Dirac delta in the time-continuous case – the Kronecker delta *can* be realised for numerical purposes. It is equivalent to applying a step forcing and taking the difference:

$$h_\Psi[n] = \Delta\langle \hat{\Psi}_\sqcap \rangle[n] - \Delta\langle \hat{\Psi}_\sqcap \rangle[n-1]. \qquad (12)$$

This method was used in (Lucarini et al., 2017). Such external forcings we will refer to as (system) identification forcing.

When facing the practical situation of having *finite* time series, $f[l]$ and $h_\Psi[l]$, $l = 0, \ldots, L-1$, Eq. (5) of the Appendix can be used to determine the response $h_\Psi * f[l]$, $l = 0, \ldots, L-1$ (whose usefulness is coming from efficient algorithms for evaluating the discrete Fourier transform, DFT; we evaluate the DFT using Matlab's `fft`). To this end one can *pad* $f[l]$ and $h_\Psi[l]$ by $L-1$ zeros in *front* (although mind footnote 11 of the Appendix); we will denote these padded sequences by e.g. $\tilde{f}[l]$, $l = 0, \ldots, 2(L-1)$. The first useful 'half' ($l = 0, \ldots, L-2$) of the circular convolution $\text{DFT}^{-1}\{\text{DFT}\{\tilde{h}_\Psi\}\text{DFT}\{\tilde{f}\}\}$ resulting from eq. (5) will then match the linear convolution $h_\Psi * f[l]$, $l = 0, \ldots, L-1$. Unlike this calculation in frequency-domain, the calculation in time-domain using Eq. (11) is straightforward.

## 2.2 Obtaining the Green's function

First, in order to predict the response (to first order), we need to obtain e.g. the (first order) Green's function. As Eq. (4) suggests it is 'coded in' the autonomous system. A direct evaluation of this formula is, however, prone to failure (Lucarini et al., 2017). Second, we note that in practice we can study only a discrete time version of the system. This prompts that for a direct way of determining the Green's function, instead of Eq. (3) we have to use Eq. (11) (leading to Eq. (12)). It also means that we cannot infer the response characteristic of the system just by observing its autonomous dynamics, but we need to force it externally in a suitable way. Third, a single experiment regarding a chosen observable $\Psi$ is insufficient to make, but an ensemble of experiments (appropriately initialised) is needed to obtain the expected value $\langle \hat{\Psi} \rangle$ (notation introduced in Sec. 2.1, first appearing in Eq. (11)). Clearly, only a finite number of experiments is feasible to run, so we obtain an approximation of $\langle \hat{\Psi} \rangle$, where the error is some correlated noise process. This correlation can be negligible with an infrequent sampling allowed by, say, a slow forcing to be applied.[3] We use the data that was used for (Lucarini et al., 2017), which consist of some ensembles

---

[3] As noted in Sec. 2.1, the approximation $\langle \Psi \rangle^{(1)}[n] \approx \langle \hat{\Psi} \rangle^{(1)}[n]$ – even with infinite ensemble size – is the better the better the forcing $f$ is approximated by a staircase function with a certain sampling time $T$. Therefore, the larger $T$, the worse the approximation, and the more white as a noise the error with



of 200 members, and we have produced new data belonging to new forcing scenarios, to be described in Sec. 2.4, that consist of ensembles of 20 members.

As already spelled out in Sec. 2.1, two identification forcing types are particularly suitable to determine the Green's function; one is a step forcing, and the other is the Kronecker delta. When a random statistical error is present due to the finite ensemble

size, represented say by a Gaussian random variable $\xi$, it is actually *better to use a Kronecker delta* forcing for the following reason. Using the step forcing one needs to take the difference of consecutive values – what is sometimes called 'differencing' – of the response sequence (12). This way at any time the variance of the error is that of the *difference* of two random variables, $\xi_1$ and $\xi_2$, both distributed identically to the original random variable $\xi$. For Gaussian variables it is straightforward to show that $\mathrm{Var}[\xi_1 - \xi_2] = 2\mathrm{Var}[\xi]$. Note that we assume that $\xi$ is the same random variable to a good approximation under the delta

and step forcings.[4] Nevertheless, we apply a step forcing also in our new experiments, so that we are able to make use of data produced for (Lucarini et al., 2017) in a consistent manner. Examples of the response to step forcings are displayed in Fig. 2. The similarity of the responses to greenhouse and solar forcings here, and so the Green's functions, is consistent with the findings of (Merlis et al., 2014; Hansen et al.).

It is important to appreciate the following trade-off. For a better signal-to-noise ratio one can apply a more powerful identifi-

cation forcing. However, in the case of the presence of nonlinearities, the more powerful the forcing signal the larger the error in estimating the Green's function *belonging to the base state* $\langle \Psi \rangle_0$ (even without noise).

We make here two more comments on the issue with noise. First, instead of instantaneous samples of the observable $\Psi$ and the corresponding Green's function, we will consider, like in (Lucarini et al., 2017), *annual averages*, $\bar{\Psi}[n] = \int_0^1 d\nu \Psi((n + \nu)T)$. This is sensible given the slow rate of change that the applied forcing represents; and it also greatly reduces the noise

level. In this regard we point out that annual averages too obey Eq. (11) *exactly* if the forcing is constant over a year, because the order of summations can be interchanged, whereby a well-defined DT Green's function belonging to the annual average emerges. We will use only annually constant staircase-like forcings in our experiments (Sec. 2.4, ), so that it be clear that a linear prediction of the response has an error not because Eq. (11) does not apply exactly, but because of the missing higher order perturbative terms appearing in (2). Second, the said enhancement of noise by differencing in (12) cannot be

overcome by working in frequency-domain. The Green's function, via frequency-domain applying Eq. (1), is expressed as $h_\Psi = \mathrm{DTFT}^{-1}\{\mathrm{DTFT}\{\Delta\langle\hat{\Psi}_\Gamma\rangle\}/\mathrm{DTFT}\{f_\Gamma\}\}$, where $1/\mathrm{DTFT}\{f_\Gamma\} = 1 - e^{-i\omega}$. The latter is the very factor arising in the DTFT of a differenced sequence. The only way that we are aware of to avoid the differencing and thereby reducing the noise is that by using a Kronecker delta identification forcing as argued above[5].

---

a finite ensemble size. However, it is not the whiteness of this noise is what matters but its magnitude, so there is not really an "accuracy vs whiteness" trade-off situation regarding the choice of $T$. However, shortly we discuss how a trade-off situation does arise regarding the choice of $T$ concerning indeed the *magnitude* of the noise.

[4]Clearly, when the noise-like fluctuation is a genuine part of the response, the variance of these fluctuations are not the same under the two said types of the forcing.

[5]There exist filtering techniques, but they introduce some assumptions either on the functional form of the Green's function (parametric techniques), or on the goodness of fit (nonparametric techniques) of their estimate to, say, one of the described straightforward (noisy) estimates (such as a minimal root-mean-square-error). One can use e.g. Matlab's `impulseest`.



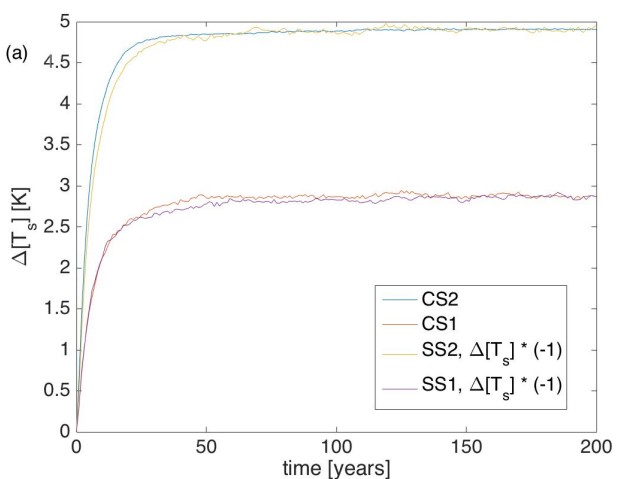
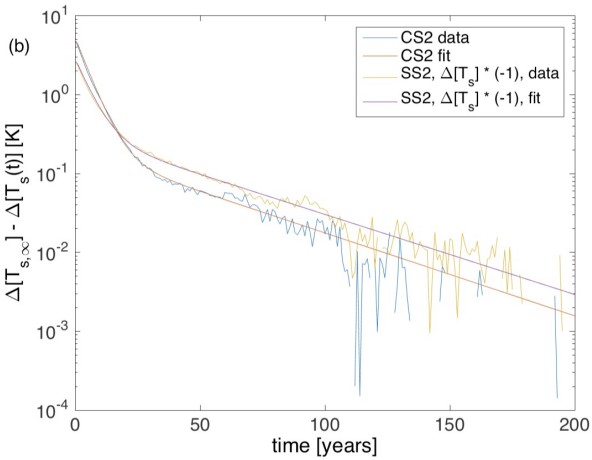

**Figure 2.** Simulated response to step forcings. The chosen observable is the global average surface air temperature $[T_s]$. The identification forcing scenarios are those of CS2, CS1, SS2, SS1 from Table 1. (a) After a subtraction of the limit value and displaying the response on lin-log scales (b), it is revealed that the high-dimensional system behaves very much like a noise-driven linear 2-box model, also called a vector autoregressive (VAR) model, in view of the considered global scale variable. The two time scales of the VAR models fitted to the CS2 and SS2 data are about 5 and 40 years. Note: the angle brackets denoting ensemble average are dropped from diagram annotations throughout the paper.

## 2.3 The inverse problem

When different forcings act in the same time, their first-order contributions to the response – as discussed in Sec. 1.1 – can be *superimposed*. Hence, when we desire a certain total response $\Delta\langle\Psi_\Sigma\rangle(t)$ to a *combined forcing* when all forcings are given but one, there is a *unique* form of that one required to fulfill our desire. In terms of the geoengineering problem of our interest (Sec. 1.2), the required solar forcing $f_s$ in order to achieve a total response $\Delta\langle\Psi_\Sigma\rangle$ given a greenhouse forcing $f_g$ can be *expressed*, to a first-order approximation, in frequency-domain as stated under (9). With the most obvious choice of *cancellation*, $\Delta\langle\Psi_\Sigma\rangle = 0$, Eq. (9) simplifies to:

$$f_s(\omega) \approx -\frac{\chi_{\Psi,g}(\omega)}{\chi_{\Psi,s}(\omega)} f_g(\omega). \tag{13}$$

Note that the forcings are defined by (1) to have zero reference values belonging to $\langle\Psi_\Sigma\rangle_0$, and so there is no need to write $\Delta$ in their notation. However, in practice when finite time series are available, the simplification is not so trivial. As described in the end of Sec. 2.1, in place of the FT's we have to calculate in Eq. (9) with DFT$\{\tilde{f}_g\}$, DFT$\{\tilde{h}_{\Psi,s}\}$ and DFT$\{\tilde{h}_{\Psi,g}\}$. *Furthermore,* the DFT in place of $\Delta\langle\Psi_\Sigma\rangle(\omega)$ is that of a sequence $\Delta\langle\check{\Psi}_\Sigma\rangle[l]$, only the first useful 'half' ($l = 0, \dots, L-2$) of which is zero, as dictated by our requirements, but its second half ($l = L-1, \dots, 2(L-1)$) has nonzero values in general. These nonzero values characterize the total response to combined *step* forcings (to do with the 'gap' mentioned in the caption of Fig. 3), but





also depend to a certain extent on the particular finite $f_g[l]$ presented. The reason for this is that the Green's function is given only up to a finite time, which becomes clear upon inspection of the workings of the convolution of finite time series. The said nonzero values are given of course by

$$\Delta\langle\check{\Psi}_\Sigma\rangle = \mathrm{DFT}^{-1}\{\mathrm{DFT}\{\tilde{h}_{\Psi,g}\}\mathrm{DFT}\{\tilde{f}_g\} + \mathrm{DFT}\{\tilde{h}_{\Psi,s}\}\mathrm{DFT}\{\tilde{f}_s\}\}, \tag{14}$$

where, however, $\tilde{f}_s$ is not known being the sought-for object. The idea is that we can look for $\tilde{f}_s$ by an *iterative* procedure, which is initialised, say, by $\tilde{f}_s = \tilde{f}_g$. Note that if $h_{\Psi,g}$ and $h_{\Psi,s}$ are not dissimilar, nor are $f_g$ and $f_s$; that is, the initial value is not far from the solution, which gives hope that it is within the basin of attraction to the solution. In each iterate we

1. evaluate Eq. (14) using a current estimate of $\tilde{f}_s$, but replacing beforehand any nonzeros in the first half of that $\tilde{f}_s$ by zeros;

2. in the resulting $\Delta\langle\check{\Psi}_\Sigma\rangle[l]$ we replace any nonzeros in the *first* half by zeros in order to have it in the right form; and then

3. we get a new estimate for $\tilde{f}_s$ using a formula analogous with Eq. (9).

Ideally, the first half of the $\tilde{f}_s$ estimates in stage 3. converge to zero, and the second half to some nontrivial form that is the solution. In our experience (results not shown) this is the case for systems with fairly simple and smoothly varying Green's functions. However, when the same Green's functions are corrupted by noise, our experience is that the procedure does not necessarily converge, but iterates of $\tilde{f}_s$ can develop increasingly large and in fact regular harmonic-looking oscillatory features. It is possible to achieve convergence for some smaller but nonzero noise level. However, even then the limit function retains small oscillatory features over the full length of $\tilde{f}_s$.

We emphasize that the iterative procedure was needed because we could not predict the second nonuseful half of $\Delta\langle\check{\Psi}_\Sigma\rangle[l]$ since we do not have the Green's functions in full but with a cutoff in time. This means that by running longer and longer *ensemble* simulations, by which we can determine the Green's functions further and further in time, the solution can be approximated by a *non*iterative procedure better and better. This is clearly a numerically more expensive solution.

Working in time domain, alternatively, the inverse problem leads to performing a *deconvolution*:

$$f_s = (\Delta\langle\check{\Psi}_\Sigma\rangle - h_{\Psi,g} * f_g) *^{-1} h_{\Psi,s}. \tag{15}$$

Note that we wrote $\Delta\langle\check{\Psi}_\Sigma\rangle[l]$ in the above which should exactly correspond to the appropriately defined circular convolution in (14) as $l = 0,\dots,2(L-1)$. Clearly, in time domain too $f_s[l]$, $l = 0,\dots,L-1$, is obtained iteratively, in three stages similarly as outlined above in frequency domain. One can use Matlab's `deconv` to perform the deconvolution. We find in simple examples studied (results not shown) that without noise the procedure in time domain leads to the very same solution as the procedure in frequency domain. This is not the case with additive noise, which means that the deconvolution/inverse problem is *ill-posed* in this case. However, the weaker the noise, the closer the outcome to the true solution, either in time or frequency domain, as long as the procedure converges. We find that in time domain the procedure always converges to some solution, however, with increasing noise strength this solution features oscillations of increasing amplitudes as time advances. Nevertheless, for





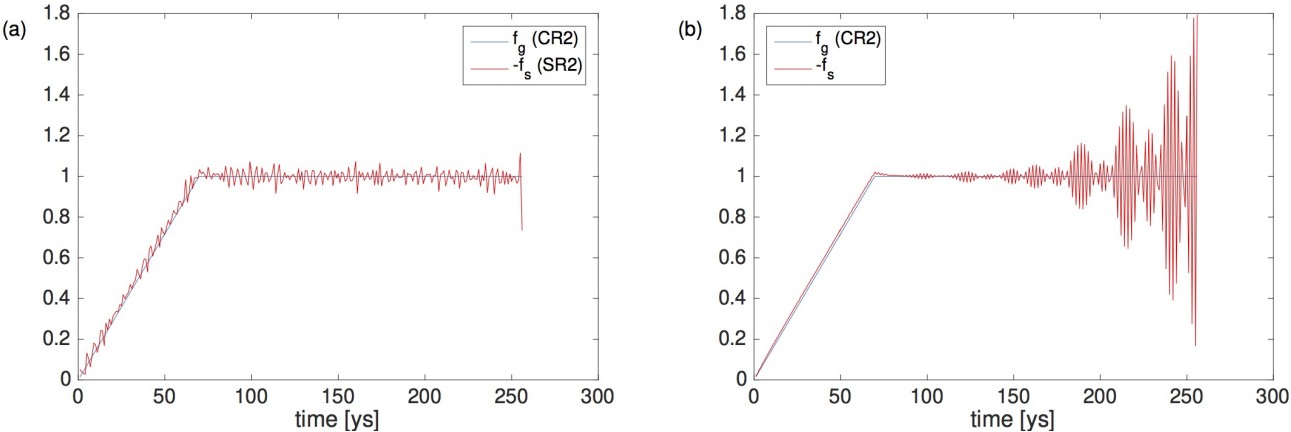

**Figure 3.** Imposed $[CO_2]$ or greenhouse forcing and required solar forcing that cancels out global average surface air temperature change. They are *normalised* for the displaying to have a unit plateau level. The required solar forcing is determined in both frequency (a) and time domains (b). We indicate in the legend which data set from Table 1 the forcings belong to. We note that in either case we *neglected the iteration*, skipping stages 1. and 2. and setting $\Delta\langle\check{\Psi}_\Sigma\rangle[l] = \Delta\langle[\check{T}_s]_\Sigma\rangle[l] = 0$ for *all l* straightaway in stage 3., the validity of which is prompted by the very similar Green's functions $h_{[T_s],g}$ and $h_{[T_s],s}$ as indicated by Fig. 2. Correspondingly, the required $f_s$ is very similar to the given $f_g$. A small gap between the red and blue ramps that can be resolved only with a smooth estimate, i.e., in panel (b) but not in (a), which gap develops quickly from the beginning of the ramps, informs us that the system responds slightly faster to the greenhouse forcing, which is already prompted by Fig. 2 (b) and the exact results (not given) of the parameter estimation by fitting. One could think that this should have to do with the fact that the greenhouse and solar forcings exert their effects on the different atmospheric layers with respect to the vertical direction differently, and the response times associated to those layers vary. However, results presented in Sec. 4 prompt that it is more to do with nonlinearity, which makes the response towards negative and positive anomalies "asymmetric".

a certain noise strength when the frequency domain procedure also converges, we find that the solution in time domain is smoother and so closer to the true solution *earlier* in time. This is also what we find considering the PlaSim data, as shown in Fig. 3. We conclude, therefore, that *it is preferable to work in time domain* using Eq. (15) to produce numerical results. Nevertheless, we will carry out our calculations in frequency domain, using e.g. the forcing signals shown in Fig. 3 (a), in

5    order to make the point that even a rough forcing signal convolved with a rough Green's function produces a not so rough response, as we will see in Sec. 3.1.1.

## 2.4    Forcing scenarios

The form of the forcing signal $f_g$ due to changes in the $[CO_2]$ concentration for which we want to solve the geoengineering inverse problem is a ramp that was used in (Lucarini et al., 2017). This is a standard forcing type, also used for the

10    CMIP6 DECK (Diagnostic, Evaluation and Characterization of Klima) protocols (Good et al., 2016). More precisely, it is not a time-continuous ramp for the reason detailed in Sec. 2.2, but the $[CO_2]$, and so $f_g$, is kept constant for one year after each



incremental increase. The $[CO_2][n+1] - [CO_2][n]$ increment is a (small) *fraction* of the current value $[CO_2][n]$, and therefore increasing in a superlinear fashion with time $[n]$, but, due to the logarithmic dependence of the radiative forcing on the $[CO_2]$ concentration (Huang and Bani Shahabadi, 2014), it realises a linear radiative forcing signal[6] $f_g[n]$, i.e., a constant-in-time $(n)$ radiative forcing increment $f_g[n+1] - f_g[n]$. Hence the naming 'ramp'. Such a form of the (radiative) forcing signal is useful in diagnosing or interpreting results. For example, if the response characteristic to solar forcing $f_s$ is similar to that of $f_g$, then the required solar forcing to cancel global change would also be approximately linear, ramp-like.

Note, however, that a linearity of the response characteristic to any forcing is checked by a comparison of the linear prediction with the truth in terms of a model simulation subject to the same forcing. Beside the nonlinearity, another factor that gives rise to a discrepancy is a statistical error due to the finite ensemble size. However, the latter has a very distinct feature that can be visually told apart easily from the contribution of nonlinearity. We reiterate that by applying a staircase-like forcing we guarantee that the said discrepancy has no contribution due to performing calculations in discrete-time.

We point out that at asymptotic times there is no discrepancy because of the way we estimate the Green's function (Sec. 2.2); the discrepancy emerges *transiently* only. The all-time maximum of it is a useful intuitive measure of nonlinearity in the examined regime. However, clearly, the larger the response the larger the nonlinear contribution to it, and so – in the context of system identification – the more inaccurate our estimate of the susceptibilities (Sec. 2.2) become. Therefore, beside our *base scenario* of (overall) doubling $[CO_2]$, we will also check if we can obtain a more accurate (and so useful for the geoengineering problem) estimate of the Green's function using a *weaker* identification forcing, in particular one that results in *half* of the (overall) radiative forcing change of that by doubling $[CO_2]$ (realised by $[CO_2]_\infty/[CO_2]_0 = \sqrt{2}$, according to the above mentioned logarithmic law (Huang and Bani Shahabadi, 2014)). Note that in the case of this weaker forcing, irrespective of the different plateau level, the increments of the $[CO_2]$ changes realise the same 1%/yr relative change.

We refer the reader to Table 1 for an overview of the various identification and test forcing scenarios that we used in the present study. Among them we have CQ2 defined by 0.1%/yr relative changes, which makes it a much slower change than the base scenario. The response to such a slow ramp forcing should be ramp-like as long as the linear term in (2) dominates over the nonlinear ones. This forcing scenario will therefore provide us another reference in interpreting other results with respect to linearity.

In the said table we did not indicate the plateau level of the solar forcing $f_s$ used in conjunction with $f_g$. We chose this level such that the response asymptotically in terms of the global average surface air temperature is the same but of opposite sign as that due to the corresponding $f_g$. This level can be easily determined to a good approximation by an iterative procedure. Beside those in Table 1, we will introduce a few more forcing scenarios in Sec. 4 that will aid the interpretation of our results and others that give improved results.

---

[6] This is meant to be in a loose sense, because strictly speaking the realised radiative greenhouse forcing (which we do not even try to define here) must not be considered as an external forcing. The external forcing is the $[CO_2]$ concentration indeed. A logarithmic scaling of this signal, however, makes no difference insomuch as a causal Green's functions exist between this scaled variable and well-behaved observables. The scaling is intuitive and standard practice, and we will allow ourselves to refer to $\ln([CO_2]/[CO_2]_0)$ as the radiative greenhouse forcing.





**Table 1.** Sets of simulation data specified by the forcing. Each data set is codenamed by a three character code; the first character coding the quantity in which the forcing is presented (C for $[CO_2]$, S for 'solar irradiance'); the second character coding the 'form' of the forcing signal (S for 'step', R for 'ramp'; Q for 'slow ramp'); and the third character coding the plateau level of the (corresponding – see main text) greenhouse forcing (2 for $[CO_2]_\infty/[CO_2]_0 = 2$, and 1 for $[CO_2]_\infty/[CO_2]_0 = \sqrt{2}$). The CS2 and CR2 data sets are preexisting to the present study (Lucarini et al., 2017) containing 200 ensemble members. All new data sets listed here contain 20 ensemble members each, except for CQ2 which contains 10.

|         | Step |            | Ramp |            | Slow ramp | Form    |
|---------|------|------------|------|------------|-----------|---------|
| Forcing | 2    | $\sqrt{2}$ | 2    | $\sqrt{2}$ | 2         | Plateau |
| $[CO_2]$ | CS2 | CS1        | CR2  | CR1        | CQ2       |         |
| Solar   | SS2  | SS1        | SR2  | SR1        |           |         |
| Combined |     |            | BR2  | BR1        |           |         |
| Quantity |     |            |      |            |           |         |

## 3 Results

### 3.1 Surface air temperature

#### 3.1.1 Global average

This is the variable with respect to which we *prescribe* the cancellation. We do *not* consider any other variable in this role
throughout the present study. Having predicted the solar forcings (SR1, SR2) required to produce no total response used in
combination with prescribed $[CO_2]$ forcings (CR1, CR2) adopting the methodology described in Sec. 2.3 (see also the note in
the figure caption of Fig. 3), we plot the predicted linear responses in Fig. 4 (a). Clearly, these predictions can be viewed either
as components of the *predicted* total response (BR1, BR2), or the predicted response in separate scenarios (CR1, CR2, SR1,
SR2). Alongside these predictions we plot the true response in the scenarios when the forcings are applied separately, i.e., the
responses evaluated by direct numerical simulations (CR1, CR2, SR1, SR2). Regarding our objective (I), the comparison of
prediction and truth reveals that (i) the response to stronger forcing is more nonlinear in the case of greenhouse forcing (CR2)
in comparison with solar forcing (SR2); and that (ii) with a weaker identification (CS1, SS1) and test forcing (CR1, SR1) the
linear prediction for CR1 is much better than for CR2, while SR1 is seemingly as good as SR2. For the scenarios of combined
forcing (BR1, BR2) only the true response is nontrivial if nonlinear, which is displayed in Fig. 4 (b). Indeed, because of the
nonlinearity, the total asymptotic response is nonzero. It is visibly nonzero even with the weaker forcings. However, it is just
about 10% of that with greenhouse forcing solely even in the case of the stronger forcings.

The pronounced nonlinearity (i) shows up also in other experiments. With a very slow forcing CQ2 we registered the response
as shown in Fig. 5. Despite that the rate of forcing is unchanged throughout the almost 700 years, the response switches to a





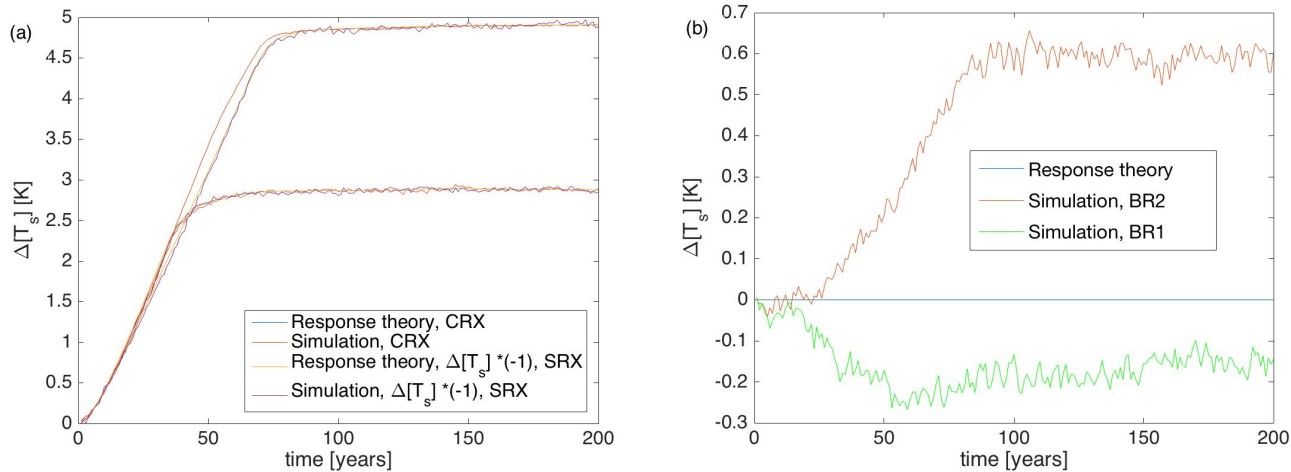

**Figure 4.** Predicted and true surface air temperature responses to ramp-like forcings. Forcing scenarios are: (a) CR1, CR2, SR1, SR2, (b) BR1, BR2. Note that in panel (a) the two yellow curves perfectly cover the corresponding blue ones, because $f_s$ is calculated to cancel global warming at all times.

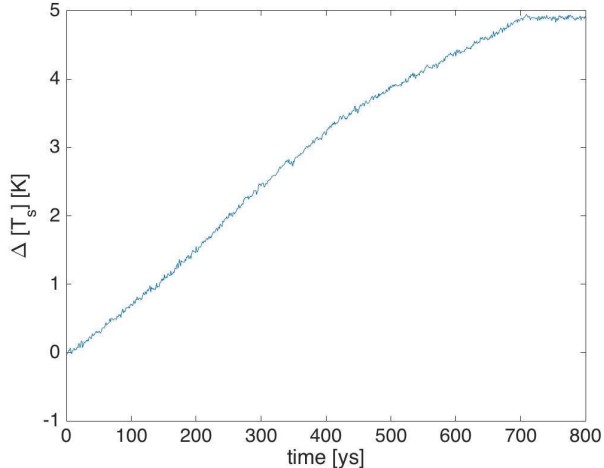

**Figure 5.** True response of the global mean surface temperature under a very slow ramp forcing, CQ2.




slower rate between 400 to 500 years, or, between 3 to 4 [K] changes in the temperature[7]. The placement of this change of the rate, compared to the asymptotic temperature change of almost 3 K upon the weaker CR1 forcing seen in panel (a), is in good agreement with the observation of a much more closely linear response to that weaker forcing as compared to CR2. A crude indicator of non/linearity can be extracted from the CQ2 experiment, but also from comparing the asymptotic/stationary responses (denoted by a subscript $\infty$) in the XX1 and XX2 experiments as follows:

$$(\Delta\langle[T_s]_{\infty,2}\rangle/\Delta\langle[T_s]_{\infty,1}\rangle)/(f_{\infty,2}/f_{\infty,1}). \tag{16}$$

(Note that we write an 'X' in place of one of the possible characters in the scenario identification code when it does not matter which of the possible characters is written there.) This value is 0.99 with solar forcing and 0.85 with greenhouse forcing, in agreement with what the comparison of predicted and true responses seen in Fig. 4 (a) allowed us to conclude above.

### 3.1.2 Zonal average

Regarding our objective (II), we begin with the zonal average surface air temperature for our *diagnosis* of any residual total response in terms of other observables than the one for which a desired evolution has been (attempted to be) enforced. First, we show results with the $\sqrt{2}$-fold $[CO_2]$ increase (CR1, BR1). Treating zonal means, following (Lucarini et al., 2017) (where only the case of $[CO_2]$-doubling was treated), in a similar fashion to global means informs us that the response to either greenhouse or solar forcing is the strongest at high-latitude/polar regions; see Figs. 6 (a) and (c). Unsurprisingly, this is where the response is most nonlinear, as indicated by Figs. 6 (b) and (d), showing the difference between truth and prediction. In these diagrams we see colors for nonzero values also in the whole stretch of stationary forcing, however, for the different latitudes separately, after a fast approach of the stationary climate, the time-average should be zero by means of the used methodology (except for a small finite data statistical error). As a consequence of the said nonlinearities, in the high-latitude regions linear response theory 'badly fails' to predict the total response to combined forcing, also in the regime of stationary climate; compare Figs. 7 (a) and (b) showing the prediction and truth, respectively. The latter statement pertains to our objective (I), assessing the applicability of linear response theory, beside that here the *choice of the observable* to analyse pertains to objective (II).

In addition to such a visual comparison it is customary to quantify the discrepancy by measuring the error of prediction *relative* to the true value. However, the true value can be zero at certain latitudes which makes this naive relative error measure lacking an obvious meaning. In these situations it is customary (Tornqvist et al., 1985) to analyse the following relative error:

$$e_1 = \frac{|\Delta\langle\Psi\rangle_{\mathrm{BRX}} - \langle\Psi\rangle_{\mathrm{BRX}}^{(1)}|}{|\Delta\langle\Psi\rangle_{\mathrm{BRX}}| + |\langle\Psi\rangle_{\mathrm{BRX}}^{(1)}|}. \tag{17}$$

It takes on values from [0,1] for all values of $\Delta\langle\Psi\rangle_{\mathrm{BRX}}$ and $\langle\Psi\rangle_{\mathrm{BRX}}^{(1)}$; and, clearly, a larger value should be considered worse. We note that in Eq. (17) $\langle\Psi\rangle_{\mathrm{BRX}}^{(1)}$ is meant to be an estimator of the actual quantity, which estimator is biased, but for keeping it simple, we do not introduce a separate symbol for the estimator. Another possibility in our situation is measuring the error

---

[7]Clearly, a slower rate of change of the response to a slow forcing translates to a smaller static susceptibility (at $\omega = 0$), i.e., sensitivity.

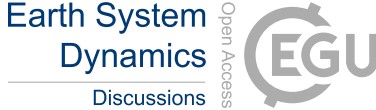



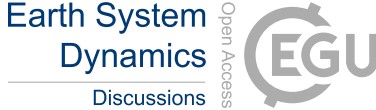

**Figure 6.** Response of the zonally-averaged surface air temperature to ramp forcings. The first column shows the true responses and the second one the errors of the linear predictions. The first and second rows belong to the CR1 and SR1 forcing scenarios, respectively. Similar diagrams as in the first row but for CR2 are shown in Fig. 6 of (Lucarini et al., 2017).





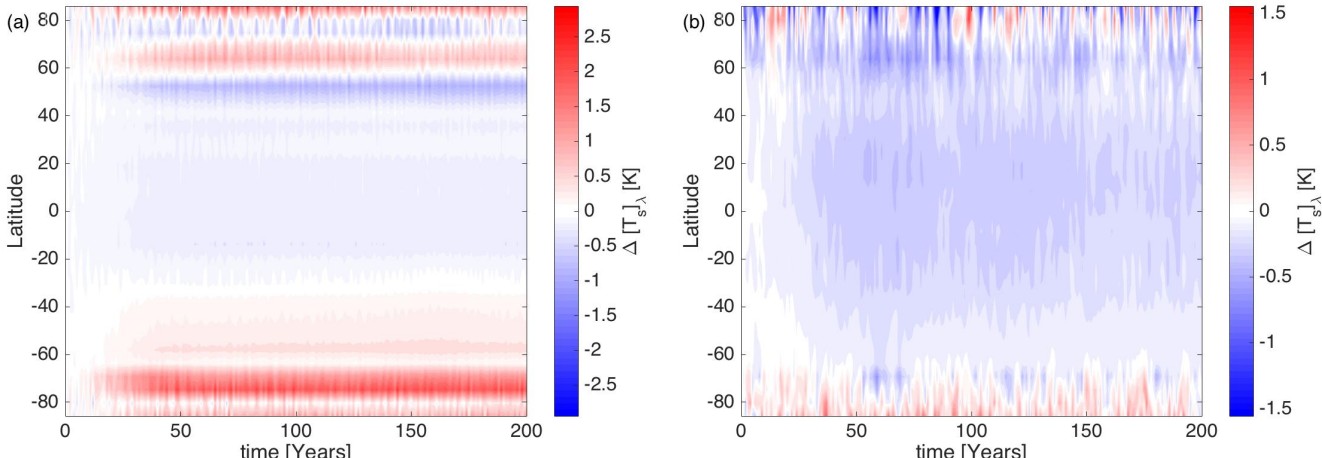

**Figure 7.** Predicted (a) and true (b) total responses of the zonally-averaged surface air temperature to combined ramp forcings (BR1).

of prediction of the response to combined forcing relative to the response to one of the forcings:

$$e_2 = \frac{|\Delta\langle\Psi\rangle_{\mathrm{BRX}} - \langle\Psi\rangle_{\mathrm{BRX}}^{(1)}|}{\Delta\langle\Psi\rangle_{\mathrm{CRX}}}. \qquad (18)$$

$e_2$ shares the property of $e_1$ that it is decreasing (perhaps not monotonically) with a decreasing stationary $[CO_2]_\infty$ level. We evaluate $e_1$ and $e_2$ only with respect to the stationary climate, in which case the estimation is very accurate as we can take an average also with respect to time. Fig. 8 (a) shows the result in the case of the weaker forcing (CR1, BR1). Both $e_1$ and $e_2$ indicate that the prediction is the poorest at some high-latitude regions.

With $[CO_2]$-doubling (CR2, BR2), results shown in Fig. 8 (b), the performance has a different characteristic as compared with weak forcing. Both $e_1$ and $e_2$ are the highest at both equatorial and some high-latitude regions, and somewhat less at polar and some Southern Hemisphere midlatide regions.

### 3.1.3 Spatial pattern

A more comprehensive view of the spatial variation of the response (II) is given by the distribution over the 2D surface, predicting or 'measuring' (computing) the response in each gridpoint separately, as done in (Lucarini et al., 2017). Similarly to zonal averages, the response patterns to greenhouse and solar forcings are very similar in the stationary climate regimes; see Figs. 9 (a) and (b) for the strong forcings CR2 and SR2, respectively. These patterns are misaligned slightly, which results in nonzero predicted total responses of opposite sign in neighbouring regions, BR2. It is shown in panel (c) of the same Figure. The picture for the weaker forcings, CR1, SR1 (not shown), BR1 (Fig.9 (e)), is similar.

Unsurprisingly, large predicted residual total responses occur where the response is large to either greenhouse or solar forcing alone. However, the predicted total response turns out to be grossly erroneous (I); the truth regarding the surface air



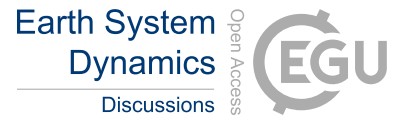

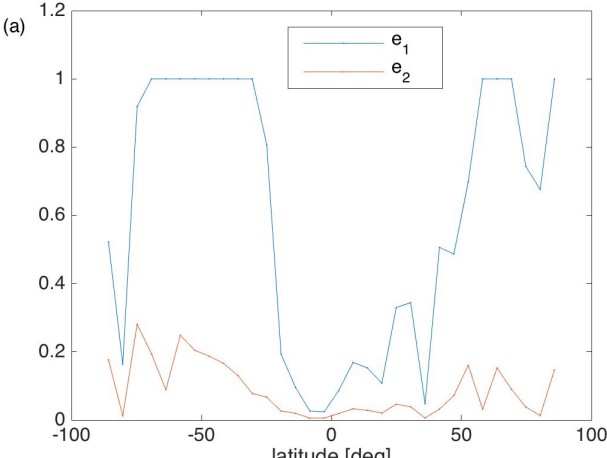
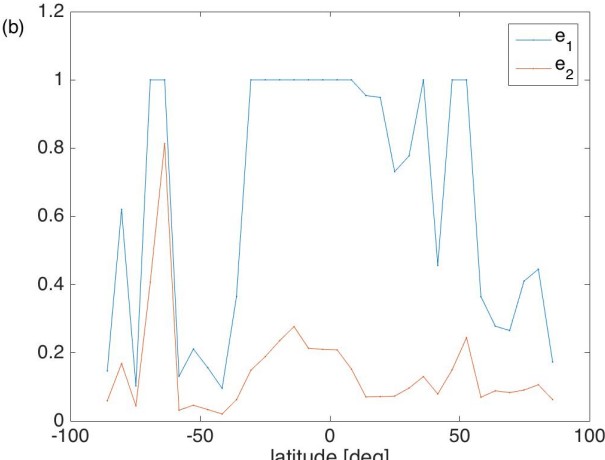

**Figure 8.** Relative errors $e_1$ and $e_2$ defined respectively by Eqs. (17) and (18) for the predicted total responses of the zonally-averaged surface air temperature to combined ramp forcings. (a) is a companion diagram to those in Figs. 6 and 7 belonging to the weak forcing scenarios (CR1, BR1), whereas (b) shows the same for the stronger forcing scenarios (CR2, BR2). Discrete data points are connected by lines to aid reading the diagram.

temperature, shown in panel (d) for BR2 and (f) for BR1, is much 'better behaved' for both forcing strengths: *significant cancellation is achieved even locally*. (We note that the overwhelmingly red (blue) color in panel (d) ((f)) is consistent with the signs of the true residual total global change shown in Fig. 4 (b).) However, looking at the temperatures at the highest model level (II), nearest the tropopause, the response under combined forcing (BX2) relative to the response under, say, solar forcing

alone (SX2) is much larger at the tropopause – evidenced in Fig. 10 (a) and (b) – in comparison with the surface, the latter given by comparison of Fig. 9 (b) and (f).

## 3.2 Annual precipitation

Here we present results for another diagnostic observable (II) the annual precipitation $P_y$ with a reversed order with respect to the spatial characteristics as compared to Sec 3.1; and we do not distribute the material into subsections. In terms of the

10 spatial patterns of response, very similar conclusions can be drawn for the precipitation as for the surface air temperature, which is supported by the set of diagrams in Fig. 11. Only that the largest responses are observed at equatorial regions. Most importantly: *significant cancellation is actually achieved as opposed to the 'damning' linear prediction*. This is so even if the solar forcing used is the same as before, i.e., that was determined with the aim to cancel global warming (not wettening). This clearly suggests that the response characteristic of $P_y$ to greenhouse and solar forcing, say in terms of the respective Green's

functions, are very similar, similarly to the corresponding Green's functions of $[T_s]$. Nevertheless, a difference of the response characteristics of $P_y$ and $[T_s]$ is manifested in the nonzero linear prediction for the total response in the stationary climate seen





**Figure 9.** Spatial variation of the stationary climate in terms of the surface air temperature belonging to different forcing levels specified by plateaus of forcings collected in Table 1. (a) CX2 (b) SX2 (c) BX2 (d) BX2 (e) BX1 (f) BX1. All diagrams picture the truth, except for (c) and (e) which show the linear predictions. Mind the different ranges of the temperature for the colourbars.



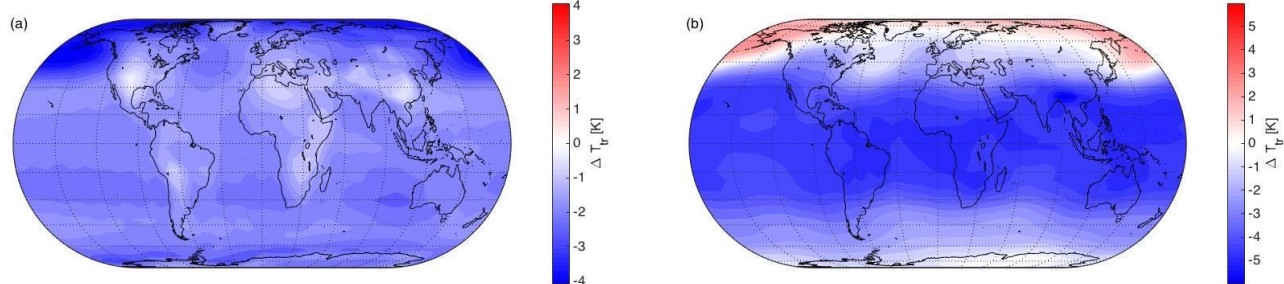

**Figure 10.** True spatial variation of the stationary climate in terms of the air temperature in the topmost model layer, nearest to the tropopause. (a) BX2 (b) SX2.

**Table 2.** Global average stationary climatology of the annual precipitation belonging to different forcing levels.

| Forcing | CX1 | CX2 | SX1 | SX2 |
|---|---|---|---|---|
| $\Delta[P_y]_\infty$ [mm] | 74 | 124 | -71 | -121 |

in Fig. 12. In comparison with the true total responses plotted in the same diagram, the linear prediction is quite 'unreliable', as can be expected from what we have seen for the spatial patterns. Otherwise, both the predicted and the true total responses to combined forcing look rather negligible to the responses to the greenhouse or solar forcings acting separately, listed in Table 2. Interestingly, the transient responses (not shown) have similar qualities to those of the temperature: nonlinearity is most

obvious for CR2 as opposed to CR1, SR1, SR2.

We note that Equatorial drying under a similar geoengineering scenario has also been reported in (Ferraro et al., 2014). However, in that study a quadrupling of [CO$_2$] was considered. We point out that it does seem to matter what levels of change we consider: under [CO$_2$]-doubling we find significantly, and not proportionately (in terms of the greehouse forcing), more true (not what is predicted) drying than in the case of the $\sqrt{2}$-fold [CO$_2$] increase. This finding of nonproportionate drying can,

however, have different reasons. One candidate is that the response under combined forcing is nonlinear; and the other one is that (assuming that the response under combined forcing is approximately linear) the required solar forcing was determined inaccurately (which resulted already in a residual response as seen in Fig. 4 (b)). We emphasize that results for the surface air temperature in this regard, to be presented in the following, do not inform us about the situation with the precipitation, and we will not attempt to reach a conclusion about it.  Drying while global average surface temperature would be maintained in a

model was reported also in (Ricke et al., 2010, 2012).



**Figure 11.** Same as Fig. 9 but for the annual precipitation.



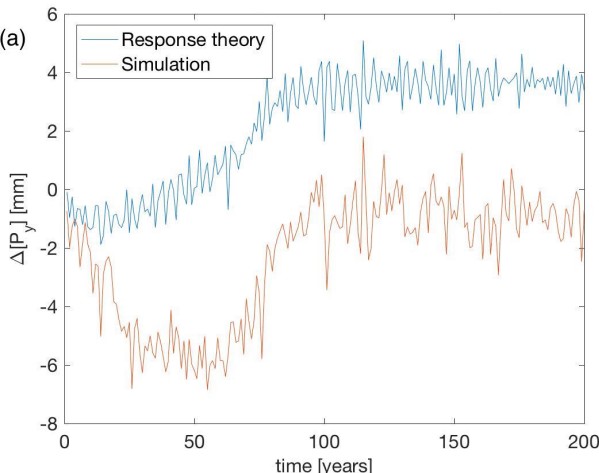 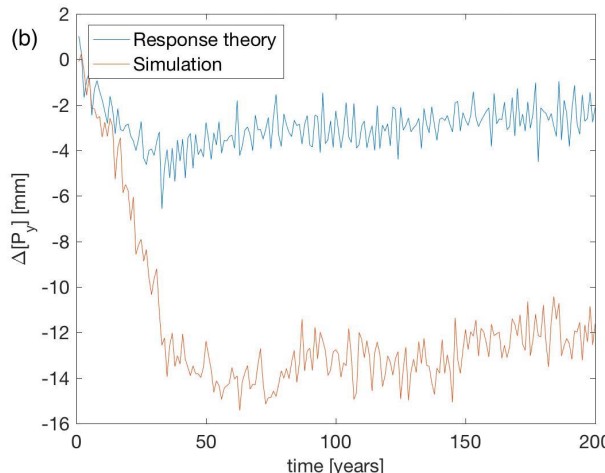

**Figure 12.** Same as Fig. 4 but for the annual precipitation.

## 4 Improved methodology and results

The very close resemblance of the patterns seen in Fig. 9 (a) and (b) hints that the effect of a changing $[CO_2]$ on the radiative forcing shaping the surface air temperature is very similar to that by a changing solar strength. However, by this data we are not properly informed about just how similar, because e.g. the CR2 and SR2 forcings act in *opposite* directions, and because

of nonlinearities they do not have to have the same effect even if the effect due to forcings acting in the same direction were indistinguishable. Therefore, we produced just that missing simulation: complimenting SS2, for which the applied solar forcing is a step of equal magnitude but opposite sign. The so-called "equilibrium climate sensitivity"[8] (ECS) is shown in Fig. 13 (a) for this forcing, to be referred to as SS2I. It is virtually indistinguishable from the pattern resulting for CS2, seen in Fig. 9 (a), including a lack of such misalignment like the comparison of panels (a) and (b) of that figure revealed. This goes beyond

the report on the (approximate) "equivalence" of greenhouse and solar forcings with respect to (asymptotic in time) *global average* surface temperature (Boschi et al., 2013), i.e., that its response is a function of the *difference* of those forcings (loosely speaking about a radiative greenhouse forcing; see footnote 6); this is extended now to *regional averages*, i.e., spatial patterns, of that variable with a remarkable degree of approximation, to be indicated by Fig. 14 (a).

The superposition of the ECSs for SS2 and SS2I, displayed in Fig. 13 (b), is in turn almost indistinguishable from the

15 asymptotic total response to combined BR2 forcing, seen in Fig. 9 (c). This pattern turns out, as explained next, to be created by even-order nonlinear perturbative terms of the response (2). The selection of the even order terms takes exactly the superposition of the responses from two experiments where the forcing is equal and has opposite sign. A general form of the problem can be posed as follows. The response to what forcings do we need to superimpose in order to eliminate terms up to order $I$? For

---

[8]Although the term 'steady state climate sensitivity' would be rather correct, as under stationary forcing the complex system is not in equilibrium but possesses a chaotic attractor.)





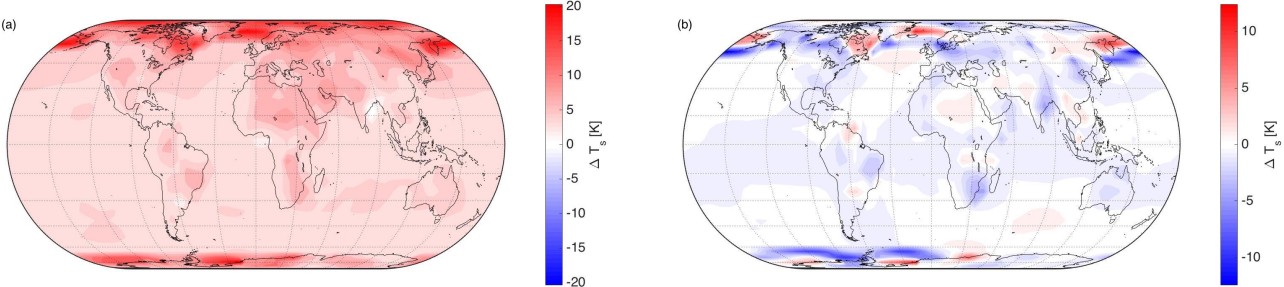

**Figure 13.** Spatial variation of the stationary climate in terms of the air temperature. (a) True response under SS2I, (b) predicted response under combined forcings used for SS2 and SS2I amounting to no forcing.

this problem the different forcings we specify as given by the same form (most simply by a step if we are only interested in the ECS) multiplied by different factors $\epsilon_i$, $i = 1, 2, \ldots$. From Eq. (2) it is straightforward to derive that these factors are the solutions of the following system of equations:

$$\sum_{i=1}^{I+1} \epsilon_i^j = 0, \quad j = 1, \ldots, I, \tag{19}$$

setting $\epsilon_1 = 1$ without loss of generality. That is, we have a number $I$ of equations for that many sought-for variables. In the case of $I = 1$ the solution is rather trivial: $\epsilon_2 = -\epsilon_1 = -1$, as we hinted already; and all the odd terms are eliminated, not only the linear one. In the case of $I = 2$ it is straightforward to show that there is no solution (the determinant of the solution formula being always negative). We have also tried to solve Eq. (19) for several $I > 2$ numerically, but without success. Therefore, we suspect that the general formulation of the problem is not 'constructive'.

In the special case of $I = 1$, instead of eliminating the even-order terms by superposition as explained, of course we can retain only the odd-order terms by subtraction. We proceed in this direction assuming that the third and higher-odd-order terms have a negligible contribution. This way we attempt to improve on the results for the linear susceptibility – and so ultimately on our prediction of the required solar forcing needed for canceling global warming. This is done clearly to the end of making an advance regarding our objective (I). We can then apply this forcing in a new experiment coded as BR2C ('C' for 'cancel').

For this experiment we can utilise (although we will not examine the transient[9]) our finding that the response characteristics to greenhouse and solar, i.e., short-wave and long-wave radiative, forcings are very similar, which would allow for applying a solar forcing that is a simple straight ramp, just like $\log([CO_2]/[CO_2]_0)(t)$, having the same length before the plateau. That is, what we improve on here is only the *level* of the plateau. It is rather straightforward to obtain the following equations for this

---

[9]The precise treatment of the transient proceeds by solving the same inverse problem as outlined in Sec. 2.3, centred around eq. (14), only that the impulse responses in that equation, e.g. $\tilde{h}_{\Psi,g}$, need to be produced as an average from two simulations each, as also done in (Gritsun and Lucarini, 2017).



level $f_{\infty,BR2C,s}$:

$$\chi_{[T_s],\infty,s} = \frac{|\Delta[T_s]_{\infty,SS2}| + |\Delta[T_s]_{\infty,SS2I}|}{2|f_{\infty,SS2}|}, \tag{20}$$

$$\chi_{[T_s],\infty,g} = \frac{|\Delta[T_s]_{\infty,CS2}| + |\Delta[T_s]_{\infty,CS2I}|}{2|f_{\infty,CS2}|}, \tag{21}$$

$$|\Delta[T_s]_{\infty,BR2C}| = \chi_{[T_s],\infty,s}|f_{\infty,BR2C,s}| - \chi_{[T_s],\infty,g}|f_{\infty,BR2C,g}|, \tag{22}$$

$$|\Delta[T_s]_{\infty,BR2C}| = 0. \tag{23}$$

The subscripts of $\infty$ refer to the asymptotic/stationary climate regime, other subscripts refer to the experiment/forcing scenario. Observe that data from a new experiment is needed, CS2I, where the 'I' indicates an experiment related with CS2 analogously to the relation of SS2I with SS2. Since we are interested in the stationary climate regime only, due to ergodicity we can produce just a single long trajectory instead of an ensemble. The result of this is $\Delta[T_s]_{\infty,CS2I} = -5.11$ [K] (while we already

have $\Delta[T_s]_{\infty,SS2I} = 4.36$ [K], and from Fig. 2 that $\Delta[T_s]_{\infty,SS2} = -\Delta[T_s]_{\infty,CS2} = -4.90$ [K]). Having that $|f_{\infty,BR2C,g}| = |f_{\infty,CS2}|$, we can express the sought-for forcing in relative terms based on the temperature data only, such as:

$$\frac{|f_{\infty,BR2C,s}|}{|f_{\infty,SS2}|} = \frac{|\Delta[T_s]_{\infty,CS2}| + |\Delta[T_s]_{\infty,CS2I}|}{|\Delta[T_s]_{\infty,SS2}| + |\Delta[T_s]_{\infty,SS2I}|} = 1.08. \tag{24}$$

In fact, we carried out the BR2C experiment independently: *iteratively* determining a solar forcing that cancels to a very good approximation the total response (similarly how the level for e.g. SS2 was determined observing the result of CS2). This forcing

in the above relative terms was found to be 1.11, agreeing well with our prediction of 1.08.

Given that our prediction is smaller than the actually needed forcing for cancellation, we can predict an *upper bound* on the actual total response to our predicted forcing by substituting into Eq. (22) the actually needed value $|f_{\infty,BR2C,s}|/|f_{\infty,SS2}| = 1.11$. This gives $\Delta[T_s]_{\infty,BR2C} < 0.134$ [K]. Considering that the total residual response with the original methodology (Sec. 2) was 0.6 [K], this means that with the improved methodology we managed to reduce the total response – considered to be

the error of prediction – almost to the *one fifth* or even less of the said first result. (Of course, the exact reduction can be easily obtained by an extra simulation, which we have not run.) In fact, some residual total response even with the improved method could be expected, as the simple measure of nonlinearity (16) indicated that linearity is much more 'violated' by increasing radiative forcing as opposed to a reducing one. This prompts that the third-order *odd* perturbative term is not 'minuscule' relative to the second order one – contrary to the assumption of our improved methodology.

Even if we managed to achieve a perfect cancellation in terms of the global averages, amounting to a success in terms of our objective (I), it is the *foundational claim* of this article that it would be still important to examine the total response in terms of any other observables regarding which the cancellation is not enforced, whether there is any unwanted residual, which is our objective (II). To this end we look at the BR2C data. In particular, in Fig. 14 we show the spatial variations of the stationary climate in terms of (a) the surface air temperature and (b) annual precipitation. The former one looks like a 'crossover' of Fig.

9 (d) and (f), and the latter like that of Fig. 11 (d) and (f). More precisely, the new diagrams look to lie in between the respective said old diagrams in the sense of an interpolation. This implies that the (true/simulated) variances with respect to space for BR2C ('perfect job'), both for temperature and precipitation, are about the same as those for BR2 ('less than perfect job'),





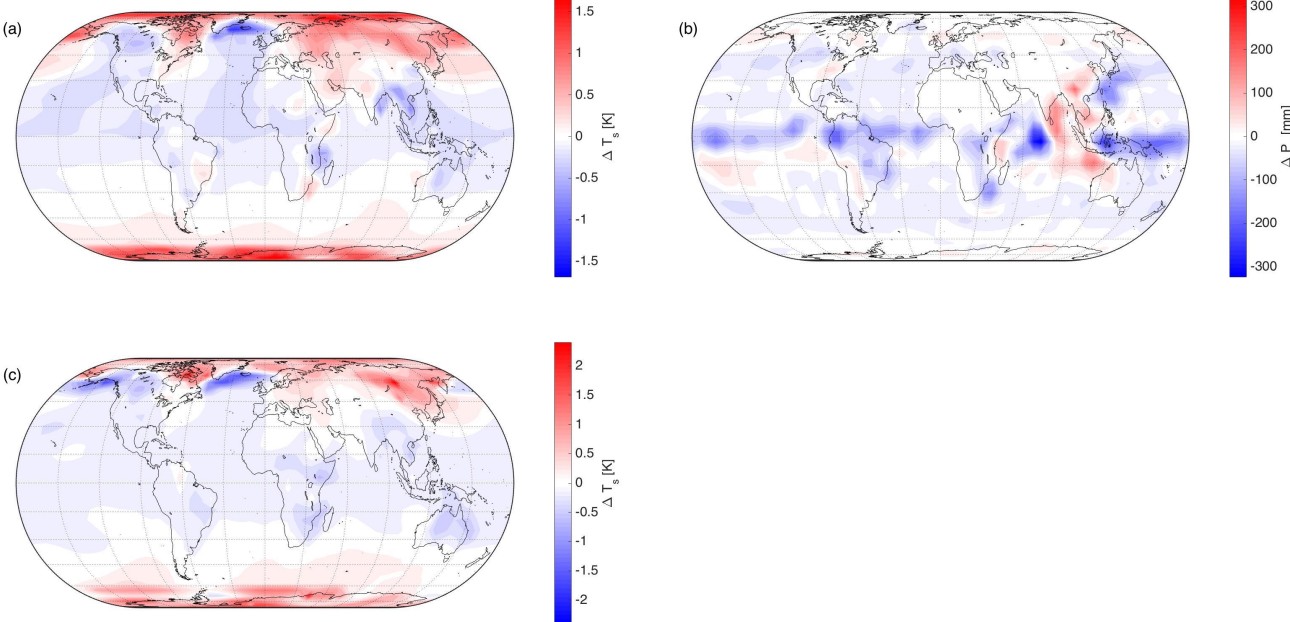

**Figure 14.** Spatial variation of the stationary climate in terms of (a) the surface air temperature and (b) annual precipitation in the BR2C experiment, when a change in the global average surface air temperature is canceled. (c) The improved linear prediction corresponding to (a).

and are much larger than the residual total responses in terms of the respective global averages for BR2. The reason for this is clearly that the response characteristics[10] to greenhouse and solar forcing coinciding with respect to the individual spatial locales are somewhat different. However, it is not really the constancy of the spatial variance with (slightly) varying levels of the applied solar forcing that is important from a practical point of view, but rather the sensitivity of the response in any locale.

5   Comparing the BR2 and BR2C scenarios, we see that the difference in terms of the climatic surface air temperature could be as much as 2 [K], which is about 10% of the maximal response under the corresponding greenhouse forcing alone.

From the point of view of (I), however, the improved methodology is a great success, as the linear prediction for the surface temperature shown in Fig. 14 (c) is in a quite good agreement with the truth shown in (a). This means, of course, that *under geoengineering (of the considered type at least) the total response is linear to a good approximation*, even with a doubling of

10   [$CO_2$]. This is consistent with finding e.g. Fig. 14 (a) a crossover of/interpolation between Fig. 9 (d) and (f). And, so, above we could have written indeed 'sensitivity' instead of 'response characteristics'. In this case it is *meaningful to strive to determine the linear susceptibility accurately*, unlike in the case of having to predict large responses which have considerable nonlinear

___

[10]This characteristics is certainly meant to be within the regimes of the actually realised total response. As this regime is finite, possibly significant nonlinear elements of the characteristics are included in our meaning. This is why we did not write at this point 'sensitivity' in place of 'characteristics'.



contributions. In the latter case one would have to work out the (more complicated) response formulae for the significant higher-order terms.

We would like to point out that the improved methodology working with [$CO_2$]-doubling to estimate susceptibilities (CS1, SS1) resulted in the same improvement in terms of slashing the residual response (BX2) as that by sticking with the naive

method but reducing the magnitude of the system identification forcing (CS1, SS1, BX1): about 1/5 in magnitude – about 0.134 and -0.15 K (see Fig. 4 (b)), respectively. However, the improved methodology enabled us to predict accurately using linear response theory also the spatial pattern, Fig. 14 (c), something that the naive method is not able to, even with reduced identification magnitudes; compare Fig. 9 (e) and (f). These can be viewed as two different components of our objective (I).

## 5   Summary and Outlook

We defined and solved an inverse problem to find a solar forcing that can cancel global warming that would otherwise result from a change in the greenhouse forcing. In fact, we can allow for other choices of the scalar observable, either with respect to the physical quantity, or considering e.g. local variables. The inverse problem itself was derived in the framework of linear response theory. Because of the true nonlinear characteristics of the response the degree of approximation of the solution specifically for the cancellation of global average surface air temperature depended on the method and its success of deter-

mining the linear susceptibilities or Green's functions belonging to the different forcings (I). The issue stems from the fact that for the estimation of the Green's functions we used *finite* magnitude external system identification forcings, in which case the nonlinearity of the response is already felt, while for the cancellation, i.e., *zero* total response, we would need the linear susceptibilities *exactly*. An inaccurately predicted required solar forcing leads to a nonzero residual true total response.

By a simple method, also used in (Gritsun and Lucarini, 2017), here, for determining the susceptibilities, we eliminate even-

order nonlinearities from the response in the system identification experiments. The price of this is having to run double as many simulations for system identification. In the scenario of doubling $CO_2$ concentration, by this method we could cut five-fold the unwanted actual total response arising instead of cancellation relative to the results without elimination. Furthermore, the linear prediction of spatial patterns using the improved susceptibility improved dramatically, owing to the fact that the total response under the considered type of geoengineering is indeed very closely linear.

We pointed out also that instead of step-wise system identification forcing, it is better to use a Kronecker delta forcing in order to achieve a better signal-to-noise ratio. As another gain from using a Kornecker delta forcing, the response would be much more modest in magnitude, and hence it would stay further off regimes with more significant contributions of nonlinear terms, and so the linear susceptibilites could be estimated more accurately even by the naive method – without having to run twice as many simulations to facilitate the improved method.

We note that the presented method of predicting a required solar forcing is based on Green's functions that are determined by externally forcing the system of interest. This is clearly not a method that could be put in practice in the case of the Earth system. Therefore, this is another reason, beside the unpredictability of the 21st century greenhouse forcing, why the method is suitable only for scenario analyses. However, the Green's functions might be possible to estimate without externally forcing





the system, just from an observation of unforced fluctuations. The crucial question in this regard is whether the fluctuation-dissipation theorem (Kubo, 1966; Leith, 1975) is applicable.

In the situation when a cancellation is achieved exactly, or to an arbitrary approximation in practice, which we achieved through an iterative procedure, we diagnosed the success – or the lack of it – of the geoengineering concept in terms of any

unwanted nonzero changes with respect to other observables (II). By our simple iterative procedure (the result of which is, otherwise, in very good agreement with the improved linear prediction, as said above) we examined only the case of the new *stationary* climate. The diagnosed observables included: local, or rather regional average, surface air temperature and annual precipitation. (In the case of the original naive method, we diagnosed also zonal average surface- and regional average near-tropopause air temperature.) In all of these we found well-measurable nonzero total responses, some of which might

very well count as significant in terms of leading potentially to serious societal or ecological impacts. We conclude that any geoengineering method is expected similarly to be able to deal with only a single objective concerning one chosen observable, but to have side effects with respect to other observables, simply because the response characteristics with respect to different observables are in general different.

**Appendix: The circular convolution theorem and its application**

Taking the discrete-time Fourier transform (DTFT) of Eq. (11) we have, via the convolution theorem for discrete sequences (Katznelson, 1976), a formally analogous version of Eq. (5) with the individual Fourier transforms approximated by Fourier series:

$$\langle \hat{\Psi} \rangle^{(1)}_{2\pi}(\omega) = \hat{\chi}_{\Psi,2\pi}(\omega) f_{2\pi}(\omega), \tag{1}$$

where e.g. $f_{2\pi}(\omega) = \text{DTFT}\{Tf[n]\} = \sum_{n=-\infty}^{\infty} Tf[n]e^{-i\omega n}$ and $f[n] = \text{DTFT}^{-1}\{T^{-1}f_{2\pi}(\omega)\} = \frac{1}{2\pi T}\int_{-\pi}^{\pi} d\omega f_{2\pi}(\omega)e^{i\omega n}$ with a normalised nondimensional angular frequency $\omega$. Featuring instead the dimensional frequency $f$ measured in Hertz

= [sec$^{-1}$], the forward and inverse transformation pairs are symmetrical: $f_{1/T}(f) = f_{2\pi}(2\pi fT) = \sum_{n=-\infty}^{\infty} Tf[n]e^{-i2\pi fTn}$ and $f[n] = T\int_{1/T} df f_{1/T}(f)e^{i2\pi fTn}$. The DTFT, a continuous function of the frequency $f$, is often sampled at $f = k/(NT)$, $k = 0, \dots, N-1$:

$$f_{1/T}(k/(NT)) = T\sum_{n=-\infty}^{\infty} f[n]e^{-i2\pi kn/N} = T\sum_{n=n_0}^{n_0+N} f_N[n]e^{-i2\pi kn/N} = T \times \text{DFT}\{f_N[n = n_0, \dots, n_0 + N]\}, \tag{2}$$

with any $n_0$, which yields the discrete Fourier transform (DFT) of the *finite* sequence $f_N[n]$, $n = n_0, \dots, n_0 + N$, where the

full infinite sequence $f_N[n]$, $n \in \mathbb{R}$, turns out to be $N$-periodic, since for the equivalence of the two sums under (2) it has to be in the so-called periodic summation form:

$$f_N[n] = \sum_{m=-\infty}^{\infty} f[n - mN]. \tag{3}$$

Therefore, when $f[n]$ is actually $N$-periodic, its DTFT is nonzero only at $f = k/(NT)$, $k \in \mathbb{R}$, and also periodic, such that the DFT of a single cycle of $f[n]$ is able to represent its DTFT. For such periodic sequences, to be denoted distinctively using a





subscript as $f_N[n]$, it can be proven (Katznelson, 1976) that:

$$y * f_N = \text{DTFT}^{-1}\{\text{DTFT}\{y\}\text{DTFT}\{f_N\}\} = \text{DFT}^{-1}\{\text{DFT}\{y_N\}\text{DFT}\{f_N\}\}, \tag{4}$$

with any nonperiodic sequence $y[n]$. Note that $y*f_N$ is referred to as the *circular convolution* of sequences $y[n]$ and $f[n]$. When the $y[n]$ and $f[n]$ sequences have a finite length, $n = 0, \ldots, N-1$ with any $N \geq 1$, so that e.g. $f_N[n] = f[\text{mod}(n, N)]$, their cir-

cular convolution can be shown (Katznelson, 1976) (https://uk.mathworks.com/help/signal/ug/linear-and-circular-convolution. html) to be:

$$(y * f_N)[n = 0, \ldots, N-1] = \sum_{k=0}^{N-1} y[k] f_N[n - k] = \text{DFT}^{-1}\{\text{DFT}\{y\}\text{DFT}\{f\}\}, \tag{5}$$

which equality is called the *circular convolution theorem*. It follows that when $y[n] = 0$ and $f[n] = 0$ for $n = 0, \ldots, N_f - 1$ and $N_y - 1$, respectively, then $(y * f_N)[\text{mod}(n - 1, N)] = (y * f)[n]$ for $n = N, \ldots, N + \min(N_f + N_y, N - 1)$. Furthermore,

$(y * f)[n]$, $n = 1 + N_f + N_y, \ldots, N + 1 + N_y$ is the segment that represents the part of the linear convolution that can be considered useful in the sense that it coincides with the occurrence of the finite values of $f$ in a finite time interval of length $N - N_f$. Therefore, the circular convolution $(y * f_N)[n]$ captures the useful part of the linear convolution over $n = \max(1 + N_f + N_y, N), \ldots, N + \min(1 + N_y, N_f + N_y, N - 1)$.

Therefore, when facing the practical situation of having *finite* time series, $f[l]$ and $h_\Psi[l]$, $l = 0, \ldots, L-1$, Eq. (5) can be used

to determine the response $h_\Psi * f[l]$, $l = 0, \ldots, L-1$ (whose usefulness is coming from efficient algorithms for evaluating the DFT, called fast Fourier transform algorithm, FFT). In particular, if the two sequences are to be *padded* in front by a number $N_f = N_h = N_0$ of zeros equally (so that the circular convolution (5) be well-defined), then the reconstructed length of the linear convolution $h_\Psi * f$ (the response of a causal system coinciding with the forcing) is $1 + N_0 - \max(N_0 - L + 1, 0)$. This is a linear function of $N_0$ saturating at $N_0 = L - 1$ reaching the full length $L$. Therefore, for simplicity one can pad by $N_0 = L - 1$

zeros[11], and we will *denote these padded sequences* by e.g. $\tilde{f}[l]$, $l = 0, \ldots, 2(L-1)$. Note that padding with fewer or no zeros results in a circular convolution that better approximates either the useful or the not useful part of the linear convolution, which approximation is the better the more zeros are used. In the extreme case of no padding, very little of the useful part could be well-approximated. The key to the applicability of Eq. (4) is that it does not matter how the forcing $f[n]$ – and with it the response $\langle\hat{\Psi}\rangle[n]$ – continue after our experiment, and so they can be thought of as periodic.

*Competing interests.* The authors declare to have no competing interests.

*Acknowledgements.* This work is part of the EU Horizon 2020 project CRESCENDO (under grant No. 641816); the financial support is gratefully acknowledged. It also received support from the EU Blue Action project (under grant No. 727852); and VL acknowledges support from the DFG Sfb/Transregio TRR181 project.

[11] This results in an odd sequence length, which has an adverse effect on the common fft algorithm performance. Therefore, in actual practice one can produce time series data of length $L$ being some power of 2, and pad by an equal number of zeros.



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
