# Peer review of "Critical Assessment of Geoengineering Strategies using Response Theory"

_Earth System Dynamics, 2018_

## Referee Comment (RC1) · Anonymous Referee #1 · 3 Jun 2018

The authors need to do a better job of reading the existing literature on the subject and articulating what new contribution they have; in particular see the first (and second) comment below – it looks like this paper is simply repeating previously published work but in a less clear manner and using a worse climate model. The wording also needs to be significantly improved for clarity; in particular, the paper tends to use way more mathematical notation and terminology than is necessary to describe fairly easy concepts. Finally, the paper does not use a state-of-the-art climate model, so that at best the paper could be useful for illustrating methodology (were it not for the fact that others have already done so), and not for actually useful results regarding whether linearity is or is not a useful approximation, or the degree of "cancellation" of spatial patterns of temperature or precipitation (since many other papers have already explored that over

the last 20 years in far better climate models).

1. My very first thought on the very first line of the abstract was, hasn't this already been done? Go look at MacMartin and Kravitz, in APD in 2016 doi: 10.5194/acp-16-15789-2016. The same authors (among others) have shown linearity of the response to solar geoengineering in a whole range of papers.

2. The remainder of the first paragraph sounds exactly like one element of the approach used in this paper: doi:10.1098/rsta.2014.0134.

3. Second paragraph, line 9, this seems not very well worded. The bigger comment here is that no real system is actually linear, the question is simply whether a linear approximation is good enough to make useful predictions. So a residual error doesn't mean that a linear approximation is not correct – it is never correct, just maybe useful if that error is sufficiently small. The other comment here is that the sentence is extremely hard to understand; one has to guess that you made some linear approximation to predict something that should have been zero to interpret the first half of the sentence (in general, some arbitrary combination of greenhouse and solar forcing can give you any residual you want, including zero if you want it to), and I have no clue what you mean by "linear susceptibility". (The susceptibility of what output to what input?)

4. L12-13, again, "under geoengineering" is too vague. One can pick any level of geoengineering one wants and (subject to saturation limits) get any level of global mean temperature you want.

5. L13-17, how is this different from the conclusions reached in many many dozens of previous papers on the subject? (I can't even think of a single paper to point to as it is quite well known, perhaps Kravitz et al 2013 in JGR, or even going back to Govindasamy and Caldeira in 2000 in GRL; really one could find this observation in every paper that has ever been written on the subject.) This isn't new, and as such, this isn't a contribution of your work, and thus does not belong in the abstract.

6. L15, strike the word "ideal". There's no way to justify that word here, nor is it necessary.

7. L21-22, I've never heard someone talk about "great risk" as an "enormous gain". Suspect you didn't mean that, reword.

8. L24, this is not true. See comment #1 above for an example (and presumably any paper that has cited it). See comment #2 for another example.

9. Ditto P2 L1-3.

10. Section 1.1 is poorly motivated:

(a) First, is the goal of this paper to learn something useful about geoengineering, or is the goal of this paper to show off particular mathematical tools? If it is the former, then surely some of the problem description in section 1.2 should precede this, and be used to motivate this. Furthermore, if it is the former, and the math is a means to an end, then surely one shouldn't include more mathematical notation or concepts than are actually required to solve the problem at hand, yet there is zero motivation in this section to say why these concepts are needed. If you want this paper to be read by climate scientists (or really, anyone at all other than the authors), you need more motivation. For example, if the only thing one is interested in is "the response" of system (1) to some forcing f(t), that doesn't need any of the subsequent paragraph or concepts... that's much more simply given by the solution to the differential equation.

(b) Eq. (1) as written contains nothing stochastic, so as written pretty much the rest of this entire page is superfluous to solving eq (1). (And similarly, whenever you talk about ensemble-mean quantities, which only make sense if there is some stochastic component to the problem.) If you are intending later on to introduce some stochastic component, then you should both say so before you introduce the math, and further-more justify why that math is actually needed – that is, why you think it is insufficient to simply include additive stochastic forcing. To my mind, the forced-response (i.e., ensemble mean) is the only thing we are trying to predict, and while for a nonlinear system the forced response will itself depend on the statistical properties of the stochastic forcing, I would expect this to be such a trivial effect that it would be perfectly reasonable to use eq (1) in a deterministic sense – in which case, much of the math introduced on this page is irrelevant. If you disagree, you should explain that, rather than just regurgitating math onto the page.

(c) Also, as written, there is no indication of why you have chosen to separate epsilon from g, since obviously that doesn't actually change anything. (And I didn't find anything later in the paper that explained it either...)

(d) Note that while page 2 is, as written, nearly impenetrable to your intended audience, the first paragraph on page 3 seems written at too low a level.

11. Sec 1.2, first line, choose a word other than "mitigated" which (unfortunately) has a specific narrow connotation in climate change (referring only to reduction in greenhouse gas emissions)

12. First par of Sec 1.2 should be rewritten for clarity, e.g., start by saying "here are the approaches that could do this" and then "we will only consider a solar reduction". Note that SRM refers generally to all of these methods, it does not refer specifically to a solar reduction.

13. P4, L14, again, this is not true, see point #2 above.

14. L15, even aside from the observation that other studies have indeed formulated the required input as the solution to an inverse problem, it is also not true that other studies have only considered predetermined inputs as implied by this sentence, there are now quite a few papers that have used a feedback algorithm to adjust the forcing level; one should at least insert the word "many" (i.e., "In many previous studies...").

15. L30, I think the first half of this sentence is unnecessary.

16. P5, L9-12, this seems like a deliberate and unnecessary introduction of jargon

to describe an incredibly simple concept; as implied in earlier comments, who is your intended audience? This does not seem written in a way intended to be read by any climate scientist (indeed, strikes me as deliberately written in such a way so that no climate scientist will read it).

17. L13-16, again, this is very well known to any reader of ESD, and while it is important, it is unclear to me how using a different set of terminology is helpful.

18. Regarding the footnote, the first three sentences seem completely redundant with the text before the footnote, while the last sentence is normative.

19. L20-23, a trivial editorial comment, but can you use bullets rather than hyphens that could be confused for a minus sign, and then perhaps a colon before the clarification for the same reason.

20. L27, can you translate this sentence into English? (I think you're saying something trivial.)

21. P6, L2, no, that is not an open question. Read virtually ANY paper that has ever been written about geoengineering.

22. P6, L4-5, what do you mean that "response theory can predict spatial patterns"? I think that the spatial patterns of response to forcing are a result of climate physics, and are predicted by climate models.

23. P6, L10, minor wording, but it is well known that you can't "cancel" the effects of greenhouse forcing but rather you can offset some effects or reduce the effects, or some such wording. . .

24. P7. . . given availability of simulation output from more realistic models (e.g. GeoMIP), why use this one?

25. P7, first paragraph. . . see also comment #1; this has been done before specifically for geoengineering.

26. P7, L13, I don't know what basis you could conceivably have for asserting that reality would "most likely" be worse.

27. P8, L5, what distinction are you emphasizing with the \hat notation?

28. P8, L26, why are you claiming that a single experiment is insufficient? I agree if one is trying to diagnose whether the dynamic system is linear or not, but if the system is assumed to be LTI and noiseless, then the Green's function can indeed be perfectly computed from a single experiment

29. P9, L3-10, the answer as to which is better depends on what range of frequencies one is interested in as well, and thus I think your comparison of two possible choices (out of an infinite number of possibilities) is a bit simplified (in contrast to most of the paper that seems to take an overly complicated approach to everything). I think Ben Kravitz has a paper in the last couple of years on system identification in the context of climate science.

30. P9, L13, note you missed the year (2005) in Hansen et al, though I don't recall that paper dealing with the dynamics at all (I didn't go back and look it up though). Caldeira and Myhrvold (in ERL, 2013) for CO2 and MacMynowski et al (in GRL, 2011) are the ones I might cite to say that the dynamics themselves are similar for both forcing mechanisms (in those papers, both satisfying a semi-infinite diffusion model).

31. L14-16, I'm guessing this point is also made in Ben Kravitz' paper, but you should check (looked up the reference I was thinking of, it was 2017 in ACP, but I didn't go back and re-read it to refresh my memory).

32. L18... yet again, this seems like you are planning on precisely duplicating the methodology of MacMartin and Kravitz, 2016!

33. Figure 2 caption, Caldeira and Myhrvold also did the fit with a 2-box model... as did Isaac Held a few years earlier. (The former of these papers notes that the most appropriate functional form depends on the specific climate model, though my

guess would be that it is difficult to distinguish from step-response type inputs. There's another paper in the last couple of months, also by MacMartin et al, that also uses a semi-infinite diffusion model in fitting the step response, in Phil. Trans. I think.)

34. Section 2.3, this happens to be a case where doing the analysis in the time-domain instead of the frequency domain would be utterly trivial. While the paper noted in my comment #2 used a particular functional form for the Green's function (aka impulse response), in general all one needs to do is uniquely solve for the required forcing at time 1, then given that, uniquely solve for the forcing required at time 2, and so forth. I'm not sure why you're choosing to make this complicated when there's such an obvious, easy, exact solution method available (that also, in a real situation, has the benefit of not needing to know future values for the greenhouse forcing). Seems like this section could be replaced by a sentence. . . (or, indeed, by a reference). (If the time-domain process is susceptible to noise in the estimated Green's function, seems like the solution might be some very slight smoothing – my presumption being that you are implicitly doing that with the frequency domain approach.)

35. Re Fig 3 caption, penultimate sentence, no, I wouldn't think that. . . the atmospheric response is very fast, and the time-constants that you see in annually-averaged data result (mostly) from ocean memory; there might still be differences in the time-constants due to different latitudinal patterns of forcing though.

36. P13, L6, I would be cautious about using the word "linear" here given that much of the paper involves concept of a linear dynamic system, which isn't quite the same meaning as a signal with a constant slope.

37. P14, L14, this sounds likely for geoengineering, but I don't know that it is actually clear. . . I could write down a dynamic system where nonlinearity resulted in errors during the transient but no error at all at large time. (Might point this out at eq 16 too.)

38. P14, L15, nonlinearity is only one source of error in estimating susceptibility, the other being the climate variability (and finite ensemble size)

39. L23, again, I again agree this is plausible, but it is not rigorously justifiable.

40. Section 3.1.1, I'm confused. If you take a single forcing scenario, your "predicted" and "truth" should by definition be identical, and only deviate if you estimate the Green's function from one forcing scenario and then apply it to predict the response to another; this section needs to be much clearer about what you are doing.

41. Also note that I would suggest avoiding the word "truth" here. . . just to be clear that we are talking about models and not the real world.

42. Figure 5, it is not theoretically possible to assert based on this plot alone that the response is nonlinear. One could certainly use this plot to uniquely determine a Green's function for a linear system that would perfectly capture the response; one can only identify nonlinearity when using two different time histories of forcing. . .

43. Note re any nonlinearity found here, it is hard to assess the relevance – since, of course, all you are doing is diagnosing whether a particular climate model is or is not nonlinear. There've been tons of studies looking at linearity of the response to forcing, including many specifically for geoengineered forcing (none of which you cite), showing that in many other ESMs, the response to a solar reduction is relatively linear. (See for example the reference in my first comment.)

44. P16, L15, why "unsurprising". . . whether or not the response is well-approximated with a linear one has to do with the physics, not the magnitude of the change. (In this case I might presume saturation of ice-albedo feedback as has been noted elsewhere.)

45. Fig 6, same comment as earlier; I presume you are estimating the Green's function from one simulation and using it to predict another (otherwise the error would be zero), but you don't say so.

46. P18, L13, there's a ton of papers on this. . . starting with Govindasamy and Caldeira in 2000, through to at least Kravitz et al in 2013 (looking at GeoMIP results).

47. P18, L17, you appear to really like the word "unsurprisingly", but generally speaking

every time you use it I don't think the usage is appropriate. The difference between solar and greenhouse forcing is a result of climate physics – different spatial patterns of radiative forcing in particular; it happens to be true that those patterns of radiative forcing differ most at high latitudes (where there is little sunlight), and that that is also where there is the highest response to either forcing, but the physical reason for these two is different – the residual being largest there is NOT because the response itself is largest there – so your lack of surprise is based on an incorrect assumption! I would suggest you think more carefully before choosing to not be surprised. . .

48. Table 2, might note that this is inconsistent with every single climate model simulation (in decent climate models, that is) and basic physics; a solar reduction with the same effect on global mean temperature as CO2 will have a larger effect on precipitation (due to the "fast" response to precipitation, see Andrews et al 2009, Bala et al in PNAS, numerous other papers).

49. Figure 11 (and also Figure 9), there's something really weird about this climate model; I've never seen a precipitation response that looks remotely like that. . .

50. Footnote 8 is weird. When talking about equilibria and climate sensitivity to a steady forcing, one is referring to the forced response alone (which under mild assumptions one could estimate through a sufficiently long simulation or sufficiently many ensemble member average); the resulting quantity is not itself chaotic, only the natural climate variability that is superimposed on top of it is.

51. P25, L25-28, this is a deeply disturbing "foundational" claim on which to base this paper, since it is so central to the entire study of geoengineering, and so deeply explored in every single paper that has ever been written on the subject; surely the authors are not unaware of that?

52. Section 5, first few sentences – as noted earlier, none of this is new.

53. P27, L21-22, worth noting that this is model-specific. Other models (e.g., the

GCM's used in GeoMIP) don't exhibit as much nonlinearity as yours does.

---

## Author Comment (AC1) · 6 Aug 2018

We would like to thank the anonymous referee for the thorough consideration of our work and the long list of comments and recommendations for improvement. Very importantly, we thank the referee for pointing us to many relevant papers which did indeed consider the same or similar problems, and so should be credited.

Despite the duplication (or "multiplication") of some results or methodological developments from these references, we think that our paper still features original contributions. One of the two main new contributions is that we predict/calculate the _required_ solar forcing needed for cancelling or modulating the total response in an arbitrary desired fashion. For e.g. the GeoMIP experiment G2 the solar forcing was "simply" chosen

to have the exact same ramp shape as the "nominal"/theoretical radiative forcing due to $CO_2$ concentration forcing. Presumably this is so because the people behind the project had in mind a desire of cancelling the global average surface temperature and observed that the relevant response characteristics of models to $CO_2$ and solar forcing is very similar. In contrast with this, we outline the _general_ approach to geoengineering when 1. the choice of observable to control is arbitrary, and 2. the response characteristics to a given forcing and a geoengineering forcing are dissimilar. We mark this contribution by (I) now.

We acknowledge that our work regarding the typical side effects with respect to the spatial patterns of surface air temperature and precipitation is not original. Therefore, we remove the particulars about it from the Abstract. (The label (II) is now reserved for our other main contribution, to be detailed below.)

It appears to us that the paper doi:10.1098/rsta.2014.0134 outlines a feedback control for determining the solar forcing on the fly, similarly to our reference (MacMartin et al., 2014), by the same author. Certainly, it is the relevant approach to actually practicing geoengineering, because one doesn't need to know the pertinent Green's functions of the system. However, for an _efficient scenario analysis_, feedback control is of little use, according to our claim in the submitted manuscript: "Note that under feedback control, in a scenario analysis setting, a new simulation needs to be run for each emission scenario." (Of course, there still remains the issue that the assessment of geoengineering by an "emulator" is done based on a model, whose response characteristics are not necessarily (or rather likely not – given that different models differ from one another) the same as those of Earth's, and so the practice of geoengineering would entail further risk.)

Our choice of the Planet Simulator, PlaSim, for a model to analyse is for convenience only. 1. We had at our disposal preexisting simulation data. 2. To "keep to the word of response theory" we wanted to work with the forced response, excluding – as much as possible – internal variability (IV), because IV is out of the
control of geoengineering. The correct approximation (unbiased estimate – see our reference (Drotos et al. PRE 2016)) of the forced response is a finite ensemble approximation. To our knowledge there is no freely available large ENSEMBLE simulation data for CMIP5 models forced by step functions, but up to three realisations only, as in the GeoMIP experiments (http://climate.envsci.rutgers.edu/GeoMIP/ http://climate.envsci.rutgers.edu/GeoMIP/doc/specificationsG1_G4_v1.0.pdf). Furthermore, for our analysis of linearity, we needed/wished to run other types of simulations, e.g. SXX, CX1, BR1, BR2, CQ2, CS2I, SS2I, BR2C. We did not have the resources and in-house skill to run these simulations on a CMIP5 model. We would like to point out, in particular, that even the SSX-type experiments do not seem to exist for CMIP5 models up to now; but even if there was such data, and so we could predict the required solar forcing, there is not a BRX-type data set for CMIP5 models for which the _predicted_ solar forcing is used – not a forcing that takes the same signal shape as that of the CO2 forcing (as in the G2 GeoMIP experiment).

We omitted the seasonality in the model so that the discrete-time theory, the convolution sum (11), could apply exactly (provided an infinite ensemble size, a staircase forcing and linearity, of course). Our conclusions about non/linearity rely on this. We do not know if it applies also when beside a staircase forcing component there is also a periodic one (with a time period equal to the stair length). We think that the forced response to seasonal forcing alone is strongly nonlinear, and it's not clear to us how the system responds to a combined strong periodic and weak (say) ramp forcing: whether the response of annual averages or at certain phases of the year is linear (see for reference (Drotos et al. J Clim 2015)).

Due to the lack of seasonality, e.g. to CO2-doubling in the model the response of the global average surface air temperature is more than 2x that of the average for the CMIP models. At high latitudes locally the response is "noticeably" nonlinear. And there is a significant nonlinearity of the precipitation response, too, at Equatorial regions. The nonlinearity of the temperature response should be due to albedo saturation (as the
referee suggests) and/or a nonlinear characteristics of radiation physics. These effects show up also in CMIP models (Winton, 2013; Good et al., 2015), even if the response is more moderate, likely because they are very basic effects, and so the otherwise very high complexity of the CIMP models compared to PlaSim does not set them apart in this respect. Therefore, we think that if we find that under combined forcing/geoengineering the regional response, even if very small, is nonlinear in PlaSim, there is a "good chance" that that carries over to CMIP models.

In fact this is just what our latest analysis suggests. We had a closer look of our results and what it implies, and have come to reverse our conclusion about the (approximate) linearity of the local response. This is something that the referee claimed to have been shown by many authors, by MacMartin & Kravitz (2016) among others, and so our (original) claim is a duplication. In fact (M&K 2016) demonstrates in Figs. 3,4 the linearity under geoengineering only for global average temperature and precipitation, and for a weaker forcing. Regarding local responses, their Fig. 7 is actually inconclusive – contrary to the referee's claim (which inconclusivity is consistent with our suggestion of possible local nonlinearity also in CMIP models). That diagram shows results averaged over nine models. (Note that in an assessment of geoengineering it is not the average of all possibilities that we are interested in but the range of possibilities and how extreme some possibilities may be.) Furthermore there is an issue that the comparison of a linear prediction and the "truth" (e.g. our Figs. 14 (a,c)) can still be inconclusive. To compliment this comparison we also evaluated another measure of nonlinearity in our new Fig. 15. Thus, we propose this finding and conclusion as another main contribution of our work; and in the revised manuscript attached we label this contribution by (II) in the Abstract and elsewhere in the text. The reversed conclusion also prompted us to change the title.

A further original contribution of our paper spins out of the main proposal said above, i.e., that the required geoengineering forcing should be calculated as a solution of an inverse problem (I). In the generic situation we cannot adopt for the geoengineering

forcing the shape of the GHG radiative forcing, determining the slope – if it's a ramp – by an iterative procedure considering the stationary climate (like our BR2C experiment or the GeoMIP G2 experiment). Instead, the inverse problem has to be fed by the Green's functions. We demonstrated the importance of determining the Green's functions accurately by a dramatic improvement of the linear prediction. (Although, the linear prediction might still not be very accurate for certain observables: local, rain.) Although the methodology that we employed was published elsewhere first (Gritsun, A. and Lucarini, V, 2017), that project evolved in parallel with ours. Furthermore, in our manuscript we pointed out that in the case of linear response under geoengineering "it is meaningful to strive to determine the linear susceptibility accurately, unlike in the case of having to predict large responses which have considerable nonlinear contributions".

We note that this method of determining the impulse response/susceptibility more accurately by eliminating even order nonlinearities in the XSX experiments can be very useful considering that it might allow for reducing the number of ensemble members to simulate. We could choose to raise the identification forcing magnitude to improve the signal-to-noise ratio, as done by (M&K 2016). The increased inaccuracy of the estimate due to possible nonlinearity can then be compensated by our technique.

Next we respond to some specific points of the referee. The numbers below correspond to the numbers of the respective points of the referee's comments. Any comment that we do not address explicitly, we accepted, and catered for in the attached revised version of the manuscript. This is not the final version of the manuscript intended potentially for resubmission but a working document for the purpose of discussion; we anticipate further streamlining and rewording. Changes of any significance are highlighted by boldface typeset.

10. (a) A paper on geoengineering would best begin with the problem description. However, our aim is to frame the geoengineering problem as an inverse problem, labelled by (I), and that cannot be done in Sec. 1.2. without some introduction of the

mathematics. As we propose point (I) as a main contribution of our work, we think that it is important to explain it carefully. A crude problem description is already given in the Abstract. Some additional text now before Sec. 1.1 hopefully gives sufficient motivation. We refer to the work of MacMartin and Kravitz (2016) whose motivation was also to create an emulator based on response theory, which emulator can make predictions, or consider what-if scenarios, for an efficient assessment of geoengineering.

10. (b) We are puzzled by the referee's statements here, it does not resonate with any of our understanding of the problem at hand. Eq. (1) is a very high dimensional nonlinear system of equations. It describes the motions of a turbulent fluid. Clearly, its solution is chaotic, with a large variability on many time scales. That is, the internal climate variability is not represented by stochasticity, but deterministic processes. (For a discussion on the forced response and internal variability, see (Drotos et al. J. Clim. 2015).) We never introduce stochasticity in this work. Yet, the ensemble average can behave very simply, even linearly under weak forcing. This is a very nontrivial result of response theory. It appears that the referee thinks that Eq. (1) is linear, representing the forced response of the climate system, and the internal variability can be somehow added as random (not deterministic) noise, i.e., stochasticity. We guess that the referee has in mind LTI theory (https://en.wikipedia.org/wiki/Linear_time-invariant_theory), but response theory (see e.g. the book by Risken, or Ruelle's work) is generic in its scope, applying to any Axiom A (nice) NONlinear system.

10. (c) The concept of _linear_ response arises via a perturbation theory approach, when the full response is sought in the form of Eq. (2). The diminishing contribution of higher order terms can be represented by the powers of a small number epsilon. When one knows or assumes that the response is approximately linear, one can indeed just retain g(x). In what is now Sec. 4.1 it makes it easier – considering Eq. (2) – to see how the even-order nonlinear terms can be eliminated given that even powers of eps=-1 are the same as that of eps=1.

14. We are not aware of publications that frame geoengineering as an inverse problem.

Please kindly provide us reference. In the following paragraph we are referring to feedback control, which clarifies that we didn't mean that in all previous studies the SRM was prescribed. Nevertheless, we replaced "previous studies" by "e.g.". Also, we would not like to refer to "many previous studies" when we cite only one or two papers.

15. This is meant to be an emphasis on the fact that may be global warming can be cancelled by geoengineering but some climate change should still be expected.

16. Why would we not want climate scientists to read our paper? The term "isoline" should be familiar to every single climate scientist as it is a basic concept in theormo-dynamics, and climate models involve a great deal of thermodynamics. Furthermore, our exposition of geoengineering allows for a schematic seen in Fig. 1 that should aid rather than confuse understanding. We do not think this is a convoluted way of thinking, but acknowledge that it is not everyone's way.

17. We accept that this is not a new idea (although our original submission citing (Ferraro et al. 2014; Ricke et al. 2010, 2012) had already acknowledged it, even if we had not been aware of other references that the referee pointed us to), but we can certainly expect that it will be to some of our readers, and we choose to expose this in a way that is the most appealing to us, and we would like to believe that we won't be alone with this.

18. We do not think that there is a redundancy here. The footnote can be considered to fall under considerations of social sciences. No normative statement is made in the footnote. Normative statements express moral judgements or wishes how things should be. The last sentence expresses a factual statement, or a kind of "prediction".

19. The same latex syntax for the "itemize" environment using the plain article doc-ument class produces a bullet. The hyphen is likely produced by the esd document class. Hopefully this issue will be fixed in the copyediting process if our manuscript makes it.

20. The sentence is grammatically correct. We suppose that the meaning of "isoline" is the key to understanding the statement, and believe that our target audience will not be overly challenged. Also, Fig. 3 of the cited paper (Boschi et al. 2013) would help the readers who look it up.

22. We think that we use the word "prediction" correctly in this context. For example, as the referee suggests, a model can provide the prediction of a future state of the system which is not _explicitly_ represented in the model, but somehow coded by the model equations. Similarly, we can use the Green's function to predict the evolution of the system under some forcing, which evolution is not explicitly represented in the Green's function. The Green's function is determined either from the model equations by eq. (4) or from a forced experiment which is different from the situations that we want to have a prediction for.

23. As far as we know the word "cancel" is an exact synonym of the word "offset". Also, we wrote that the _effect_, i.e., the response, is cancelled not the cause, i.e., not the forcing. Actually, our framing of geoengineering as an inverse problem, our point (I), does imply this. This is what sets our approach apart from previous approaches, including the G2 experiment of the GeoMIP project.

27. The sentence that incorporates Eq. (11) expresses clearly, we think, that \hat{\Psi} = \Psi for staircase forcings only. For e.g. a continuous ramp forcing the realised \Psi is somewhat different from \hat{\Psi} given by the convolution sum (in which f[n] is the same for the continuos ramp and staircase functions).

28. This goes back to point 10. (b). (M&K 2016) appreciates the need for an ensemble in the bottom right of page 15791, under point 1.

29. We agree that any forcing can be used in principle for system identification, since it's just a matter of substituting that forcing and the response to it into an expression for the susceptibility, which can be obtained by rearranging our Eq. (5). We intended to "excite" the system at all frequencies, in order to have a "balanced" signal-to-noise

ratio (SNR) for the different frequencies. The spectrum of the delta function is flat, and so it is suitable for this purpose. Unless one has a specific requirement for the SNR depending on the frequency, we cannot say what the best forcing is. We would like to consider the problem of optimal identification forcing in the future – unless it is something that has already been solved by Kravitz, as the referee suggests. We would greatly appreciate if the referee could identify the paper by Kravitz that he/she referred to; unfortunately we haven't found it by ourselves. Apart from the issue of the ideal identification forcing, in the paper we compared two possibilities that have the same result in the ideal situation of having an infinite ensemble, but one is better when the ensemble is finite.

32. Taking the annual average is a minor methodological issue, and we wouldn't refer to adopting a methodology as "duplication" the same way as some results are redundant in the literature. However, we note that we actually explain that the convolution sum (11) applies exactly to the annual average, provided that the forcing is a staircase, with steps of the length of a year.

34. It wasn't obvious to us that there is a time-marching solution method to the inverse problem. However, the referee is right; we describe in the revised manuscript briefly this method too. If only we could acknowledge gratefully the referee by name! However, we note that although the referee is right also about not needing future values of the response to determine the forcing, we do need concurrent values, and so the solution technique is not applicable in practice. It is not a great issue, however, because one can employ feedback control to approximate well the solution of the inverse problem, as we discuss this now in more detail in the manuscript.

37. We do not understand the premise of the referee's statement. We think that it might be that our basic aim was not clear to the referee, namely, the framing of geoengineering as an inverse problem (I). If the actual response is nonzero at any time, despite that we use the forcing that is the solution of the inverse problem, it can be a sign of two things only: 1. the actual response is nonlinear or 2. the susceptibility that feeds into

the inverse problem was not determined accurately.

41. We do not think there is a real danger of such a misunderstanding. Also, at the first use of the word "truth" we made it clear that we mean the outcome of a model simulation.

42. We expressed clearly that in this scenario the forcing is very slow. Therefore, what we see in Fig. 5 is the static response characteristics. If it is not a straight line, then the static characteristics is nonlinear. We also note that the Green's function cannot be extracted from this data with any precision, which goes back to the point 29: the time scales present in the Green's function are not excited by such a slow forcing. Note also that, as we already wrote in the original submission, the ratio (17) (in the revised manuscript) that expresses nonlinearity can be evaluated from the data in Fig. 5 and the knowledge of the forcing used.

43. As we wrote in our main response above, we think that our analysis of the non/linearity of _regional_ averages, motivated by the referee's comments, is an original contribution. Please kindly let us know of papers that consider the linearity of regional response and make claims about it, because we are not aware of any. It is only the linearity of the response of _global averages_ that have been looked at more closely.

45. Sec. 2.2 describes how we obtain the Green's function. We use this Green's function always to make a linear prediction of the response to any other forcing, such as the ramp in the XRX experiments. This is the premise of using response theory for making predictions in the sense of point 22. above on the first place, so it goes without saying, we believe.

47. In this instance it doesn't take a physical consideration where the superposition of the patterns results in the largest values; it is just mathematics that if the patterns to superimpose are similar but slightly misaligned, then the superposition results in the largest value where the separate items had the largest values. The referee suggests

that the misalignment is the worse where the response is largest. We cannot back this up; as far as our lack of surprise is concerned, we just assumed a generic misalignment of the two patterns.

48., 49. In PlaSim there is a fast response to precipitation of opposite sign. Please see attached the Green's function for precipitation (Fig. 1) and temperature (Fig. 2) as determined from the CS2 experiment. A source of difference from published results could also be the lack of seasonality in our simulations.

Please also note the supplement to this comment:
https://www.earth-syst-dynam-discuss.net/esd-2018-30/esd-2018-30-AC1-supplement.pdf

―――――――――――――――――――

[Figure]

**Fig. 1.** Green's function for the global average annual precipitation.

[Figure]

**Fig. 2.** Green's function for the global average surface air temperature.

**Supplement:**

**Can We Use Linear Response Theory to Assess Geoengineering Strategies?**

Tamás Bódai[1,2], Valerio Lucarini[1,2,3], and Frank Lunkeit[3]

[1]Centre for the Mathematics of Planet Earth, University of Reading, UK
[2]Department of Mathematics and Statistics, University of Reading, UK
[3]CEN, Meteorological Institute, University of Hamburg, Germany

**Correspondence:** T. Bódai (t.bodai@reading.ac.uk)

**Abstract.** Geoengineering can control only some variables but not others, resulting in side-effects. We investigate in an intermediate-complexity climate model the applicability of linear response theory to assessing a geoengineering method. **The application of response theory for the assessment methodology that we are proposing is two-fold. First, as a new approach, (I) we wish to assess only the best possible geoengineering scenario for any given circumstances. This requires**
5   **solving** the following inverse problem. A given rise in carbon dioxide concentration $[CO_2]$ would result in a global climate change with respect to an appropriate ensemble average of the surface air temperature $\Delta\langle[T_s]\rangle$. We are looking for a suitable modulation of solar forcing which can cancel out the said global change **– the only case that we will analyse here –** or modulate it in some other desired fashion. It is rather straightforward to predict this solar forcing, considering an infinite time period, by linear response theory in frequency-domain as: $f_s(\omega) = (\Delta\langle[T_s]\rangle(\omega) - \chi_g(\omega)f_g(\omega))/\chi_s(\omega)$, where the $\chi$'s are
10   linear susceptibilities; and we will spell out an iterative procedure suitable for numerical implementation that applies to finite time periods too. **Second, (II) to quantify side-effects using response theory, the response with respect to uncontreolled observables, such as regional averages $\langle T_s \rangle$, must of course be approximately linear.**

We find that under geoengineering **in the sense of (I)**, i.e. the combined greenhouse and **required** solar forcing, the response $\Delta\langle[T_s]\rangle$ asymptotically is actually not zero. **This turns out to be not due to nonlinearity of the response under geoengi-**
15   **neering, but** that the linear susceptibilities $\chi$ are not determined correctly. **The error** is in fact due to a significant quadratic nonlinearity of the response under system identification achieved by a forced experiment. This nonlinear contribution can be easily removed, which results in much better estimates of the linear susceptibility, and, in turn, in a five-fold reduction in $\Delta\langle[T_s]\rangle$ under geoengineering. **This correction improves dramatically the agreement of the spatial patterns of the predicted linear and true model responses (that are actually consistent with the findings of previous studies). However, (II)**
20   **due to the nonlinearity of the response with respect to local quantities, e.g. $\langle T_s \rangle$, even under goengineering, the linear prediction is still erroneous. We find that in the examined model nonlinearities are stronger for precipitation compared to surface air temperature.**

**1  Introduction**

Geoengineering concepts with the purpose of ameliorating climate change are receiving nowadays increasing attention (Allen et al., 2014; National Research Council, a, b) (http://www.ce-conference.org/) because of the potential for an enormous gain, namely, fixing one of the greatest societal challenges primarily of a diplomatic nature, but also because of the great risk that such an unprecedented endeavour entails. However, the body of the presently available scientific analysis, albeit increasing (Lenton and Vaughan, 2013; Ferraro et al., 2014; Kravitz et al., 2011), is yet lacking the consideration of many more crucial aspects of the problem. For example, **the study of climate change in general would clearly benefit from** response theory (Kubo, 1966; Ruelle, 2009) and the theory of nonautonomous dynamical systems (Sell, 1967a, b; Romeiras et al., 1990; Crauel and Flandoli, 1994; Crauel et al., 1997; Arnold, 1998; Kloeden and Rasmussen, 2011; Carvalho et al., 2013). These mathematical tools, although having been introduced to climate science for decades (Leith, 1975; Bell, 1980; Nicolis et al., 1985), are far from being exhausted, still finding many applications of tackling problems in climate science in general (Cionni et al., 2004; Gritsun and Branstator, 2007; Kirk-Davidoff, 2009; Majda et al., 2010; Cooper et al., 2013; Lucarini and Sarno, 2011; Ragone et al., 2016; Lucarini et al., 2017; Herein et al., 2015, 2017; Bódai and Tél, 2012; Drótos et al., 2015, 2016). **The pioneering work that applies response theory to the study and efficient assessment of geoengineering in particular is due to MacMartin and Kravitz (2016). It concerns our point (II) only, and only regarding global averages. However, the regional temperature response to radiative forcing can be nonlinear (Winton, 2013; Good et al., 2015; Lucarini et al., 2017), and so it is not clear if it can be nonlinear under geoengineering too.** In the following we summarise briefly the existing mathematical tools (Sec. 1.1), and then frame the geoengineering problem as an inverse problem (I) **and provide the context for the need of assessing geoengineering strategies (II)** (Sec. 1.2).

**1.1  Elements of response theory**

In *nonautonomous dissipative dynamical systems*, like the climate system, given in the form

$$\dot{x} = F(x) + \epsilon g(x,t) \tag{1}$$

the *response* of the system to an external forcing $f(t)$ can be *unambiguously* defined in terms of the so-called *snapshot attractor* (Romeiras et al., 1990) of the system, and the natural probability distribution or the measure $\mu(x,t)$ supported by it. Both the attractor and the measure are *unique* objects; they are defined by an *ensemble* of trajectories initialized in the *infinite* past. The time-dependence of the snapshot attractor, also called a pullback attractor (Crauel and Flandoli, 1994; Arnold, 1998; Chekroun et al., 2011), and its measure give what is often termed as the 'forced response' (https://www.gfdl.noaa.gov/blogheld/3-transient-vs-equilibrium-climate-responses/), and the 'geometrical details' of theirs at any instant describe (statistical aspects of) the *internal variability* in a conceptually sound sense (Drótos et al., 2015).

For a scalar observable $\Psi(x)$ too the (forced) response is uniquely given by a projection of the measure. **Response theory (Risken, 1996; Abramov and Majda, 2008; Ruelle, 2009) asserts that the most basic ensemble-based statistics, the**

mean $\langle\Psi\rangle(t) = \int \mathbf{d}\,x\Psi(x)\mu(xt)$ **can be decomposed into linear ($j=1$) and nonlinear ($j>1$) contributions:**

$$\Delta\langle\Psi\rangle(t) = \langle\Psi\rangle(t) - \langle\Psi\rangle_0 = \sum_{j=1}^{\infty} \epsilon^j \langle\Psi\rangle^{(j)}(t), \tag{2}$$

**where the first-order, i.e., linear, term can be obtained as:**

$$\langle\Psi\rangle^{(1)}(t) = \int \mathbf{d}\,x\Psi(x) \int_{-\infty}^{\infty} \mathbf{d}\,\tau(\exp[(t-\tau)L_F(x)][L_g(x,\tau)\bar{\mu}(x)])(x,t,\tau), \tag{3}$$

5     **where $\bar{\mu}(x)$ is the natural invariant measure/probability distribution of the autonomous system ($g=0$), and the operators are defined as $L_F\mu = -\mathbf{div}(F\mu)$ and $L_g\bar{\mu} = -\mathbf{div}(g\bar{\mu})$. In (2) $\langle\Psi\rangle_0$ is the unperturbed ($\epsilon = 0$) expectation; and the series converges only if the forcing $\epsilon g(x,t)$ is small enough. If the forcing depends on time in a multiplicative way, $g(x,t) = g(x)f(t)$, then we can write that**

$$\langle\Psi\rangle^{(1)}(t) = G_\Psi^{(1)}(t) * f(t) = \int_{-\infty}^{\infty} \mathbf{d}\,\tau G_\Psi^{(1)}(\tau)f(t-\tau), \tag{4}$$

10     **where the *Green's function* is implied by Eqs. (3,4) to be**

$$G_\Psi^{(1)}(t) = \int \mathbf{d}\,x\Psi(x)(\exp[tL_F(x)][L_g(x)\bar{\mu}(x)])(x,t). \tag{5}$$

**Note that the higher-order terms $\langle\Psi\rangle^{(j)}$ can be expressed as multiple *convolution integrals* involving multi-time Green's functions (Lucarini et al., 2017).**

The convolution integral under (4) can be *interpreted* in a way that the forcing $f(t)$ is decomposed into an infinite sequence of

15 impulses, whereby the responses of the different impulses – that can be superimposed – are all given by the Green's function, whose first nonzero values occur at the time of the corresponding impulses. Although a single such *finite* impulse does not produce a nonzero response, a *continuous* sequence apparently can. Or, a single impulse of infinite magnitude, formally a Dirac delta, can also produce a response, which is clearly the Green's function itself. If the continuous train of finite impulses all have the same unit magnitude, thereby forming a step function, formally the Heaviside step function $\Theta(t)$, the response

20 is just the integral of the Green's function. Conversely, the Green's function is the derivative of the response to a unit step function. The latter prompts a numerical way of determining the Green's function, while a Dirac delta forcing is not realisable numerically.

Taking the Fourier transform (FT) of Eq. (4) we have, via the convolution theorem (Katznelson, 1976), a response formula in frequency domain:

25     $$\langle\Psi\rangle^{(1)}(\omega) = \chi_\Psi^{(1)}(\omega)f(\omega), \tag{6}$$

where $\chi_\Psi^{(1)}(\omega) = \mathrm{FT}[G_\Psi^{(1)}(t)]$ is called the linear *susceptibility*. This equation looks more useful for practical purposes as it dictates a simple multiplication instead of evaluating a convolution integral. However, in Sec. 2.1 we explain why this is not the case, which is of course to do with the transformations between time and frequency domains.

**1.2 The geoengineering problem**

It has been proposed (National Research Council, b) that the effect of greenhouse forcing can be mitigated by applying another external forcing to the Earth system, by some geoengineering means, that has, in a way, an 'opposing' effect. There are various forcing types that can achieve this, but we will consider those **– generically refereed to as "solar-radiation management" (Ricke et al., 2010, 2012) –** that can be modeled by a modulation of the solar constant. We will call this simply the "solar forcing". Clearly, these are means that modulate the shortwave incoming radiation. Readily proposed geoengineering methods include: a fleet of reflective satellites of large Sun-facing surface area put into orbit around Earth, aerosols sprayed into the atmosphere, artificially generated clouds, etc. A modulated solar constant model represents these geoengineering scenarios with a various degree of approximation, **not necessarily a good approximation (Ferraro et al., 2014)**.

Formally, the problem involves a forced/nonautonomous system, where at least two terms contribute to the forcing. For simplicity, we consider the case of only two forcing terms, and that they are both additive:

$$\dot{x} = F(x) + \epsilon(g_g(x)f_g(t) + g_s(x)f_s(t)), \tag{7}$$

where the subscripts indicate already the physical means of the forcings; 'g' for 'greenhouse' and 's' for 'solar'. Also, it is up to us to assign a value to the "small" parameter $\epsilon$, and in order to obtain a result in the uncomplicated form of (10), we choose the same $\epsilon$ for both forcing components. **Eq. (3) implies that t**he first-order contribution $\langle \Psi_\Sigma \rangle^{(1)}(t)$ of the *total response* $\Delta\langle \Psi_\Sigma \rangle$ under combined forcing, i.e., geoengineering, can be written as the superposition of first-order contributions of respective responses to the two forcings in two separate scenarios when these forcings are acting alone:

$$\langle \Psi_\Sigma \rangle^{(1)}(t) = G_{\Psi,g}^{(1)}(t) * f_g(t) + G_{\Psi,s}^{(1)}(t) * f_s(t), \tag{8}$$

whose FT is of course

$$\langle \Psi_\Sigma \rangle^{(1)}(\omega) = \chi_{\Psi,g}(\omega)f_g(\omega) + \chi_{\Psi,s}(\omega)f_s(\omega). \tag{9}$$

Note that the nonlinear response is more complicated with multiple forcings present than a sum of multiple convolution integrals (Lucarini et al., 2017) as in the single forcing scenario.

If the 'forward' problem is the prediction of the response under a given forcing, then the *inverse* problem of 'predicting' the necessary forcing for a desired response seems to be well-defined in view of the above equations. To a linear approximation the necessary or required forcing is:

$$f_s(\omega) \approx \frac{\Delta\langle \Psi_\Sigma \rangle(\omega) - \chi_{\Psi,g}(\omega)f_g(\omega)}{\chi_{\Psi,s}(\omega)}. \tag{10}$$

For the above $\epsilon = 1$ is taken. We continue to discuss the solution of the inverse problem in Sec. 2.3, including the situation when a finite time period is considered. That situation can be interpreted as a control problem, which is in fact a rather special type of *optimal* control. This way the required forcing can be 'predetermined' which need not be updated during its application. We note that this is the first time the so-called solar-radiation management (SRM) is formulated as the solution of an inverse

problem. In **e.g.** (Ricke et al., 2010, 2012) the solar forcing was constructed on the basis of some models of how much radiative forcing a sudden change of some greenhouse gas concentration or the stratospheric optical depth would yield. In addition, a scenario ensemble of SRMs was created, and a selection of the most effective SRMs was made. **The latter assessment strategy is clearly rather inefficient and inaccurate, which would still be the case had the ensemble been generated using response theory.**

The inverse/control problem would have a 'direct' practical relevance had we got $f_g(t)$ a given, as assumed. However, this is clearly not the case; predicting the greenhouse gas emissions is an extremely complicated and rather daunting task, as it is determined among others by *social* processes, for which we do not have good models. Nevertheless, efforts are underway (https://crescendoproject.eu/research/theme-4/). The current standard practice to 'deal' with this challenge, as reflected by the IPCC reports (Allen et al., 2014), is considering half a dozen 'methodologically constructed' 21st century emission scenarios. This way, instead of climate predictions one produces so-called climate *projections* belonging to hypothetical future emission scenarios. Therefore, the solution to our inverse problem has a rather *indirect* practical relevance; we can carry out at least *scenario analyses*. The reader can find elsewhere (MacMartin et al., 2014b, c, a; Kravitz et al., 2016) the description and analysis of a *feedback* control problem of *direct* practical relevance, when the solar forcing is being determined 'on the fly' with the use of some controller, adapting to a progressing greenhouse forcing, trying to realise the desired response *approximately*. Note that under feedback control, in a scenario analysis setting, a new simulation needs to be run for each emission scenario, **making it very inefficient for an extensive assessment exercise**.

We point out that in e.g. Eq. (10) we write $\Psi$ denoting a generic observable. This means that we can *choose* a particular (scalar) observable which we desire to evolve in a particular way. With a reference to the classic term of 'global warming', in contrast with 'climate change', we will attempt to enforce the cancellation of the global average surface air temperature (Sec. 3.1). With the increasingly wide-ranging analyses of climate change scenarios, however, it is clear that 'climate change' should have a comprehensive meaning, not just a synonym for 'global warming' (Conway, 5 December 2008). In fact, physical quantities other than temperature could have a larger social or ecological impact (Allen et al., 2014). Beside the *physical type* of the observable quantity, we can have different choices with respect to the *spatial scale* of the quantity, such as local, or regional (Sec. 3.1.3), zonal (Sec. 3.1.2), global (Sec. 3.1.1), etc. averages.

Once an observable $\Psi$ is chosen to evolve in a particular way, **which determines $f_s(t)$ according to (10)**, the evolution of any other observable $\Phi$ will be *a given* – the solution of a *forward* problem formally identical to (9):

$$\langle \Phi_\Sigma \rangle^{(1)}(t) = G_{\Phi,g}^{(1)}(t) * f_g(t) + G_{\Phi,s}^{(1)}(t) * f_s(t), \tag{11}$$

with $f_s$ given, of course, by (10). Clearly, $\langle \Phi_\Sigma \rangle^{(1)}(t) \neq \langle \Psi_\Sigma \rangle^{(1)}(t)$ when $G_{\Phi,g}(t) \neq G_{\Psi,g}(t)$ and/or $G_{\Phi,s}(t) \neq G_{\Psi,s}(t)$, which is the generic case. Regarding the desire of cancellation $\Delta\langle\Psi_\Sigma\rangle = 0$, we can frame geoengineering – considering for simplicity only quasistatically slow changes $f_g(t)$ – as a confinement to the 0 isoline of $\Delta\langle\Psi_\Sigma\rangle$ over the plane of $f_g$ and $f_s$ (Lucarini, 2013). In general, this isoline is different for different observables $\Phi \neq \Psi$, that is, under linear response these straight isolones fan out of the origin of the $f_g$-$f_s$ plane. This is illustrated in Fig. 1, where the curvature of the isolines for larger values of $f_g$ and $f_s$ reflect also the more general situation of nonlinear responses. It is implied then that when the system is confined

to one isoline, it can obviously not be confined to the different isolines of other variables $\Phi_i$; that is, (unwanted) changes $\Delta\langle\Phi_{i,\Sigma}\rangle \neq 0$ will ensue. In other words: the proposed geoengineering method will provide just a partial solution at best. While one aspect of the problem is solved, other aspects can be neglected, or even changed to the worse, possibly with catastrophic consequences.[1] **A long list of studies have to date addressed the issue of side-effects; see e.g. (Ricke et al., 2010, 2012; Ferraro et al., 2014; MacMartin et al., 2014a; Kravitz et al., 2013; MacMartin and Kravitz, 2016; MacMartin et al., 2018).** This possibility is the main *motivation* of our present investigation **too, concerning in particular the question (II) if response theory can provide an efficient tool to map out and quantify accurately the various side-effects of a variety of geoengineering scenarios given a variety of emission scenarios in various Earth System Models.** Having enforced (approximately, to various degrees) a cancellation of global average surface air temperature, $\Delta\langle\Psi_\Sigma\rangle = \Delta\langle[T_{s,\Sigma}]\rangle \approx 0$, we will *diagnose* unwanted changes (total response) in terms of:

– $\Phi = [T_s]_\lambda$ – zonal (Sec. 3.1.2) and

– $\Phi = T_s$ – regional averages on the surface, and

– $\Phi = T_{tr}$ – regional averages near the troposphere/tropopause (Sec. 3.1.3), and

– $\Phi = [P_y]$ and $P_y$ – annual precipitation (Sec. 3.2).

Note that we denote spatial averaging by square brackets, subscripted by the spatial variable(s) with respect to which we average over its whole range, e.g. longitudes $\lambda$ for zonal averages, and for areal/global averaging we drop the subscripting (instead of writing e.g. $[T_s]_{\lambda,\mu}$). **Some of these observables have been considered in a number of studies (Ricke et al., 2010, 2012; Ferraro et al., 2014; Kravitz et al., 2013; MacMartin and Kravitz, 2016; MacMartin et al., 2018), and our results are mostly consistent with the published ones; however, we will also focus on whether these responses can be predicted by response theory.**

We point out that **in the Planet Simulator intermediate-complexity GCM (Fraedrich, 2012), or PlaSim,** the greenhouse and solar forcings have been found approximately "equivalent" in terms of the stationary response of the global average surface air temperature (Boschi et al., 2013) insomuch that its isolines are parallel straight lines (even if there is a curvature of the surface). This was found to be the case in rather extensive ranges of the forcings, 1200-1500 Wm$^{-2}$ and 90-2880 ppm, respectively. That is, any curvature of the blue line as shown in Fig. 1 occurs outside of the said ranges. However, **to do with geoengineering the concern is** if these forcings are equivalent in the same sense in terms of other variables too, **as discussed. We will demonstrate in PlaSim that concerning regional averages $T_s$ the correspondence of forcings is still remarkable, but there is nevertheless a residual response with a nontrivial pattern under geoengineering. Furthermore our analysis indicates that (II) this residual response is not so linear, and less so for precipitation, which goes beyond (MacMartin and**
* * *
[1]Furthermore, we note that, as it is often acknowledged, 'no-one is living under the average climate'. Although, some live closer than others. That is, while the primary problem can be solved for some, even that will not be solved for others. Therefore, the debate on climate engineering is unlikely to have less political overtone and motive than the climate debate itself.

[Figure]

**Figure 1.** A cartoon of hypothetical isolines in the plane of greenhouse and solar forcings $f_g$-$f_s$ for various observables: $\Delta\langle[T_s]\rangle = 0$ – globally averaged surface air temperature, $\Delta\langle[T_a]\rangle = 0$ – globally averaged atmospheric temperature, $\Delta\langle[T_{ss}]\rangle = 0$ – averaged sea surface temperature, $\Delta\langle[T_{nhml}]\rangle = 0$ – surface air temperature averaged on the midlatitudes of the Northern hemisphere (reproduction of Fig. 5 of (Lucarini, 2013)).

**Kravitz, 2016) where the linearity of only the global average response under geoengineering is demonstrated clearly, but the linear prediction of spatial patterns were averaged over nine models.**

This work follows (Ragone et al., 2016) and (Lucarini et al., 2017). In the latter it has been demonstrated that response theory can predict spatial patterns, which, as outlined above, is one of the type of diagnostics that we use to assess the success of the geoengineering method. In both of these works the demonstrations were carried out on PlaSim (Fraedrich, 2012), but with slightly differing setups. Here we adopt the setup of (Lucarini et al., 2017) featuring meridional ocean heat transport. The present work also builds on (Gritsun and Lucarini, 2017) adopting a simple technique to obtain a better estimate of the linear susceptibility. **Clearly, a better susceptibility estimate would be useful in making a linear prediction only if the actual response is linear. While under [CO$_2$]-doubling (Ragone et al., 2016; Lucarini et al., 2017) found a nonlinear $\Delta\langle[T_s]\rangle$ response, and so no linear prediction would be productive in that case, under geoengineering the total response is aimed to be much smaller, and so in principle the response may be linear. This is found to be the case in PlaSim approximately, and so (I) by improving the susceptibility estimates we can improve greatly on our prediction of a solar forcing $f_s(t)$ required for cancellation $\Delta\langle\Psi_\Sigma\rangle(t) = 0$.**

We point out that the examined model PlaSim is lacking many realistic features, such as e.g. seasonal forcing or a deep ocean. The former deficiency results in very large global average surface temperature responses (Ragone et al., 2016), and the latter one does not allow for long time scales, typically of the order of hundred years. However, our technique is applicable in principle also to models with such long time scales. What is more, it would handle such situations powerfully given that any time *horizon* can be imposed on the analysis, constructing *transient* responses only, without the need of running very long experiments in which a new steady climate emerges upon external forcing. What makes this possible is that the Green's function is needed to be determined up to times only up to which we want to determine the response, as indicated by Eq. (4).

We wish also to clarify that our analysis technique requires the estimation of the Green's function, which is most straight-forward to do by subjecting the system to external identification forcing (Sec. 2.2), which is clearly not possible in the case of Earth. Our analysis technique is intended rather for efficient scenario analyses in *models*, where the side-effects of interest of geoengineering can be calculated for any given emission scenario, choice of observable to control in a chosen model, using negligible computer resources. **For practicing geoengineering one would use a feedback control (MacMartin et al., 2014c, a) for which the Green's function does not need to be determined while the objective should still be achieved rather accurately (even if the response was nonlinear). This practice would clearly be a "single shot", a carefully deliberated and debated choice informed by a very extensive assessment. This is to say that the numerical efficiency concerns only the assessment not the practice of geoengineering. Of course it remains a problem that the relevant Green's functions of Earth are not known accurately and we have to rely on different models for the assessment.**

[revised manuscript text omitted]
). **MacMartin and Kravitz (2016) applied a [$CO_2$]-quadrupling (and it is a standard forcing level for geoengineering studies (Ferraro et al., 2014; Kravitz et al., 2011)), however, they determined the solar forcing for cancellation not via Green's functions (Sec. 2.3), and checked the linearity of the response only up to a forcing level lower than [$CO_2$]-doubling. Their motivation for applying the high forcing level seems to be only to be able to determine the Green's function with a better SNR given that no ensemble data is available from the GeoMIP experiments.**

We make here two more comments on the issue with noise. First, instead of instantaneous samples of the observable $\Psi$ and the corresponding Green's function, we will consider, like in (Lucarini et al., 2017), *annual averages*, $\bar{\Psi}[n] = \int_0^1 d\nu \Psi((n + \nu)T)$. This is sensible given the slow rate of change that the applied forcing represents; and it also greatly reduces the noise level. In this regard we point out that annual averages too obey Eq. (12) *exactly* if the forcing is constant over a year, because the order of summations can be interchanged, whereby a well-defined DT Green's function belonging to the annual average emerges. We will use only annually constant staircase-like forcings in our experiments (Sec. 2.4, ), so that it be clear that
* * *
[3]As noted in Sec. 2.1, the approximation $\langle \Psi \rangle^{(1)}[n] \approx \langle \hat{\Psi} \rangle^{(1)}[n]$ – even with infinite ensemble size – is the better the better the forcing $f$ is approximated by a staircase function with a certain sampling time $T$. Therefore, the larger $T$, the worse the approximation, and the more white as a noise the error with a finite ensemble size. However, it is not the whiteness of this noise is what matters but its magnitude, so there is not really an "accuracy vs whiteness" trade-off situation regarding the choice of $T$. However, shortly we discuss how a trade-off situation does arise regarding the choice of $T$ concerning indeed the *magnitude* of the noise.

[4]Clearly, when the noise-like fluctuation is a genuine part of the response, the variance of these fluctuations are not the same under the two said types of the forcing.

[Figure]

[Figure]

**Figure 2.** Simulated response to step forcings. The chosen observable is the global average surface air temperature $[T_s]$. The identification forcing scenarios are those of CS2, CS1, SS2, SS1 from Table 1. (a) After a subtraction of the limit value and displaying the response on lin-log scales (b), it is revealed that the high-dimensional system behaves very much like a noise-driven linear 2-box model, also called a vector autoregressive (VAR) model, in view of the considered global scale variable, **as also recognised by MacMynowski et al. (2011); Caldeira and Myhrvold (2013)**. The two time scales of the VAR models fitted to the CS2 and SS2 data are about 5 and 40 years. **The second time scale is in a disagreement with (MacMynowski et al., 2011) and it is not clear whether a more complex model is more reliable in this respect.** 
[revised manuscript text omitted]

* * *
[6] **An anonymous referee has suggested that in time domain a simpler alternative way of obtaining the solution by a time marching procedure should exist (not relying on deconvolution). Indeed, one can break down the convolution sum (12) as** $\langle\hat{\Psi}\rangle^{(1)}[n] = \sum_{k=2}^{n} h_{\Psi,s}[k]f_s[n-k] + h_{\Psi,s}[1]f_s[n-1]$, **which can be expressed for** $f_s[n-1]$ **and consider that** $\langle\hat{\Psi}\rangle^{(1)}[n] = \sum_{k=1}^{n} h_{\Psi,g}[k]f_g[n-k]$ **is given for all** $n$. **Suppose** $f_g[0] = 0$; **then the procedure for finding** $f_s[n-1 > 0]$ **can be initialised by** $f_s[0] = 0$ **for** $n = 1$. **We have checked that it gives the same result as our procedure, reproducing the time series pattern due to a particular noise realisation in a simple example system.**

[7] This is meant to be in a loose sense, because strictly speaking the realised radiative greenhouse forcing (which we do not even try to define here) must not be considered as an external forcing. The external forcing is the $[CO_2]$ concentration indeed. A logarithmic scaling of this signal, however, makes no difference insomuch as a causal Green's functions exist between this scaled variable and well-behaved observables. The scaling is intuitive and standard practice, and we will allow ourselves to refer to $\ln([CO_2]/[CO_2]_0)$ as the radiative greenhouse forcing.

[Figure]

**Figure 3.** Imposed $[CO_2]$ or greenhouse forcing and required solar forcing that cancels out global average surface air temperature change. They are *normalised* for the displaying to have a unit plateau level. The required solar forcing is determined in both frequency (a) and time domains (b). We indicate in the legend which data set from Table 1 the forcings belong to. We note that in either case we *neglected the iteration*, skipping stages 1. and 2. and setting $\Delta\langle\check{\Psi}_\Sigma\rangle[l] = \Delta\langle[\check{T}_s]_\Sigma\rangle[l] = 0$ for *all* $l$ straightaway in stage 3., the validity of which is prompted by the very similar Green's functions $h_{[T_s],g}$ and $h_{[T_s],s}$ as indicated by Fig. 2. Correspondingly, the required $f_s$ is very similar to the given $f_g$. A small gap between the red and blue ramps that can be resolved only with a smooth estimate, i.e., in panel (b) but not in (a), which gap develops quickly from the beginning of the ramps, informs us that the system responds slightly faster to the greenhouse forcing, which is already prompted by Fig. 2 (b) and the exact results (not given) of the parameter estimation by fitting. Results presented in Sec. 4 prompt that it is **likely** to do with nonlinearity, which makes the response towards negative and positive anomalies "asymmetric", **resulting also in different spatial patterns, while the time scales associated with different locales are quite varied (not shown)**.

[revised manuscript text omitted]

15 **Lucarini et al., 2017). In Figs. 6 (b) and (d)** we see colors for nonzero values also in the whole stretch of stationary forcing, however, for the different latitudes separately, after a fast approach of the stationary climate, the time-average should be zero by means of the used methodology (except for a small finite data statistical error). As a consequence of the said nonlinearities,

[Figure]

**Figure 6.** Response of the zonally-averaged surface air temperature to ramp forcings. The first column shows the true responses and the second one the errors of the linear predictions. The first and second rows belong to the CR1 and SR1 forcing scenarios, respectively. Similar diagrams as in the first row but for CR2 are shown in Fig. 6 of (Lucarini et al., 2017).

in the high-latitude regions linear response theory 'badly fails' to predict the total response to combined forcing, also in the regime of stationary climate; compare Figs. 7 (a) and (b) showing the prediction and truth, respectively.

In addition to such a visual comparison it is customary to quantify the discrepancy by measuring the error of prediction *relative* to the true value. However, the true value can be zero at certain latitudes which makes this naive relative error measure lacking an obvious meaning. In these situations it is customary (Tornqvist et al., 1985) to analyse the following relative error:

$$e_1 = \frac{|\Delta\langle\Psi\rangle_{\text{BRX}} - \langle\Psi\rangle_{\text{BRX}}^{(1)}|}{|\Delta\langle\Psi\rangle_{\text{BRX}}| + |\langle\Psi\rangle_{\text{BRX}}^{(1)}|}. \tag{18}$$

[Figure]

**Figure 7.** Predicted (a) and true (b) total responses of the zonally-averaged surface air temperature to combined ramp forcings (BR1).

It takes on values from [0,1] for all values of $\Delta\langle\Psi\rangle_{\text{BRX}}$ and $\langle\Psi\rangle_{\text{BRX}}^{(1)}$; and, clearly, a larger value should be considered worse. **Clearly, $e_1(\mu)$ as a function of latitudes would facilitate the comparison of the predictive skill of linear response theory at different latitudes.** We note that in Eq. (18) $\langle\Psi\rangle_{\text{BRX}}^{(1)}$ is meant to be an estimator of the actual quantity, which estimator is biased, but for keeping it simple, we do not introduce a separate symbol for the estimator. Another possibility in our situation is measuring the error of prediction of the response to combined forcing relative to the response to one of the forcings:

$$e_2 = \frac{|\Delta\langle\Psi\rangle_{\text{BRX}} - \langle\Psi\rangle_{\text{BRX}}^{(1)}|}{\Delta\langle\Psi\rangle_{\text{CRX}}}. \tag{19}$$

We evaluate $e_1$ and $e_2$ only with respect to the stationary climate, in which case the estimation is very accurate as we can take an average also with respect to time. Fig. 8 (a) shows the result in the case of the weaker forcing (CR1, BR1). Both $e_1$ and $e_2$ indicate **with good agreement** that the prediction is the poorest at some high-latitude regions.

With [$CO_2$]-doubling (CR2, BR2), results shown in Fig. 8 (b), the performance has a different characteristic as compared with weak forcing. **Both $e_1$ and $e_2$ are the highest at both equatorial and some high-latitude regions**, and somewhat less at polar and some Southern Hemisphere midlatide regions.

**3.1.3 Spatial pattern**

A more comprehensive view of the spatial variation of the response is given by the distribution over the 2D surface, predicting or 'measuring' (computing) the response in each gridpoint separately, as done in (Lucarini et al., 2017). Similarly to zonal averages, the response patterns to greenhouse and solar forcings are very similar in the stationary climate regimes; see Fig. 9 (a) and (b) for the strong forcings CR2 and SR2, respectively. **(See (Hansen et al., 2005) for such a comparison in a complex model.) The patterns in Fig. 9 (a) and (b)** are misaligned slightly, which results in nonzero predicted total responses

[Figure]

[Figure]

**Figure 8.** Relative errors $e_1$ and $e_2$ defined respectively by Eqs. (18) and (19) for the predicted total responses of the zonally-averaged surface air temperature to combined ramp forcings. (a) is a companion diagram to those in Figs. 6 and 7 belonging to the weak forcing scenarios (CR1, BR1), whereas (b) shows the same for the stronger forcing scenarios (CR2, BR2). Discrete data points are connected by lines to aid reading the diagram.

of opposite sign in neighbouring regions, BR2. It is shown in panel (c) of the same Figure. The picture for the weaker forcings, CR1, SR1 (not shown), BR1 (Fig.9 (e)), is similar.

Unsurprisingly, large predicted residual total responses occur where the response is large to either greenhouse or solar forcing alone. However, the predicted total response turns out to be grossly erroneous (II); the truth regarding the surface
5   air temperature, shown in panel (d) for BR2 and (f) for BR1, is much 'better behaved' for both forcing strengths: *significant cancellation is achieved even locally*. (We note that the overwhelmingly red (blue) color in panel (d) ((f)) is consistent with the signs of the true residual total global change shown in Fig. 4 (b).) However, looking at the temperatures at the highest model level, nearest the tropopause, the response under combined forcing (BX2) relative to the response under, say, solar forcing alone (SX2) is much larger at the tropopause – evidenced in Fig. 10 (a) and (b) – in comparison with the surface, the latter
10   given by comparison of Fig. 9 (b) and (f).

**3.2 Annual precipitation**

Here we present results for another diagnostic observable the annual precipitation $P_y$ with a reversed order with respect to the spatial characteristics as compared to Sec 3.1; and we do not distribute the material into subsections. In terms of the spatial patterns of response, very similar conclusions can be drawn for the precipitation as for the surface air temperature, which is
15   supported by the set of diagrams in Fig. 11. **However,** the largest responses are observed at equatorial regions, **and it is not clear what mechanism causes it**. Most importantly: *significant cancellation is actually achieved as opposed to the 'damning'*

[Figure]

**Figure 9.** Spatial variation of the stationary climate in terms of the surface air temperature belonging to different forcing levels specified by plateaus of forcings collected in Table 1. (a) CX2 (b) SX2 (c) BX2 (d) BX2 (e) BX1 (f) BX1. All diagrams picture the truth, except for (c) and (e) which show the linear predictions. Mind the different ranges of the temperature for the colourbars.

[Figure]

**Figure 10.** True spatial variation of the stationary climate in terms of the air temperature in the topmost model layer, nearest to the tropopause. (a) BX2 (b) SX2.

**Table 2.** Global average stationary climatology of the annual precipitation belonging to different forcing levels.

| Forcing | CX1 | CX2 | SX1 | SX2 |
|---|---|---|---|---|
| $\Delta\langle[P_y]\rangle_\infty$ [mm] | 74 | 124 | -71 | -121 |

*linear prediction*. This is so even if the solar forcing used is the same as before, i.e., that was determined with the aim to cancel global warming (not wettening; **in the same spirit as Fig. 4 of (MacMartin and Kravitz, 2016)**). This clearly suggests that the response characteristic of $P_y$ to greenhouse and solar forcing, say in terms of the respective Green's functions, are very similar, similarly to the corresponding Green's functions of $T_s$. Nevertheless, a difference of the response characteristics of $[P_y]$ and
5    $[T_s]$ is manifested in the nonzero linear prediction for the total response in the stationary climate seen in Fig. 12. In comparison with the true total responses plotted in the same diagram, the linear prediction is quite 'unreliable', as can be expected from **the mismatch of the true and predicted** spatial patterns. Otherwise, both the predicted and the true total **global mean** responses to combined forcing look rather negligible to the responses to the greenhouse or solar forcings acting separately, listed in Table 2. Interestingly, the transient responses (not shown) have similar qualities to those of the temperature: nonlinearity is most
10    obvious for CR2 as opposed to CR1, SR1, SR2.

     We note that Equatorial drying under a similar geoengineering scenario has also been reported in (Ferraro et al., 2014; MacMartin and Kravitz, 2016). However, in **these studies** a quadrupling of $[CO_2]$ was considered. We point out that it does seem to matter what levels of change we consider: under $[CO_2]$-doubling we find **actually less drying** than in the case of the $\sqrt{2}$-fold $[CO_2]$ increase. This finding can, however, have different reasons. One candidate is that the response under combined
15    forcing is nonlinear; and the other one is that (assuming that the response under combined forcing is approximately linear) the required solar forcing was determined inaccurately (which resulted already in a residual response as seen in Fig. 4 (b)). **Note that in (Ferraro et al., 2014; MacMartin and Kravitz, 2016) an exact cancellation of global mean surface temperature was achieved in the stationary climate, like e.g. in the G1 GeoMIP experiment. Given this, Fig. 4 of (MacMartin and**

[Figure]

**Figure 11.** Same as Fig. 9 but for the annual precipitation.

[Figure]

**Figure 12.** Same as Fig. 4 **(b)** but for the annual precipitation, **and showing separately the cases of (a) BR2 and (b) BR1.**.

**Kravitz, 2016) indicates that the response of the global mean is approximately linear in most CMIP5 models considered, at least up to a certain forcing level that was actually lower than [CO$_2$]-doubling. In the following we show that both of these effects play a role, i.e., nonlinearity is also present in our case, however, it should not be the dominant component.** Drying while global average surface temperature would be maintained in a model was reported also in (Ricke et al., 2010, 2012).

**4 Improved methodology and results**

**4.1 Achieving cancellation (I)**

The very close resemblance of the patterns seen in Fig. 9 (a) and (b) hints that the effect of a changing [CO$_2$] on the radiative forcing shaping the surface air temperature is very similar to that by a changing solar strength. However, by this data we are not properly informed about just how similar, because e.g. the CR2 and SR2 forcings act in *opposite* directions, and because of nonlinearities they do not have to have the same effect even if the effect due to forcings acting in the same direction were indistinguishable. Therefore, we produced just that missing simulation: complimenting SS2, for which the applied solar forcing is a step of equal magnitude but opposite sign. For this forcing the stationary climate is shown in Fig. 15 (a), to be referred to as SS2I. It is virtually indistinguishable from the pattern resulting for CS2, seen in Fig. 9 (a), including a lack of such misalignment like the comparison of panels (a) and (b) of Fig. 9 revealed. This goes beyond the report on the (approximate) "equivalence" of greenhouse and solar forcings with respect to (asymptotic in time) *global average* surface temperature (Boschi et al., 2013); this is extended now to *regional averages*, i.e., spatial patterns, of that variable with a remarkable degree of approximation. (**Just how close this equivalence is** is to be indicated by Fig. 14 (a).)

[Figure]

**Figure 13.** Spatial variation of the stationary climate in terms of the air temperature. (a) True response under SS2I, (b) predicted response under combined forcings used for SS2 and SS2I amounting to no forcing.

The superposition of the stationary climates for SS2 and SS2I, displayed in Fig. 15 (b), is in turn almost indistinguishable from the asymptotic total response to combined BR2 forcing, seen in Fig. 9 (c). By inspection of Eq. (2), this pattern turns out to be created by even-order nonlinear perturbative terms of the response. The selection of the even order terms takes exactly the superposition of the responses from two experiments where the forcing is equal and has opposite sign: $\varepsilon_1 = -\varepsilon_2$.

Instead of eliminating the even-order terms by superposition, of course we can retain only the odd-order terms by subtraction. We proceed in this direction assuming that the third and higher-odd-order terms have a negligible contribution. This way we attempt to improve on the results for the linear susceptibility – and so ultimately on our prediction of the required solar forcing needed for canceling global warming. This is done clearly to the end of making an advance regarding our objective (I). We can then apply this forcing in a new experiment coded as BR2C ('C' for 'cancel'). For this experiment we can utilise (although we will not examine the transient[9]) our finding that the response characteristics to greenhouse and solar, i.e., short-wave and long-wave radiative, forcings are very similar, which would allow for applying a solar forcing that is a simple straight ramp, just like $\log([CO_2]/[CO_2]_0)(t)$, having the same length before the plateau. **(This should be the rationale behind the G2 experiments of GeoMIP.)** That is, what we improve on here is only the *level* of the plateau. It is rather straightforward to obtain the following equations for this level $f_{\infty,BR2C,s}$:

$$\chi_{[T_s],\infty,s} = \frac{|\Delta\langle[T_s]\rangle_{\infty,SS2}| + |\Delta\langle[T_s]\rangle_{\infty,SS2I}|}{2|f_{\infty,SS2}|}, \tag{20}$$

$$\chi_{[T_s],\infty,g} = \frac{|\Delta\langle[T_s]\rangle_{\infty,CS2}| + |\Delta\langle[T_s]\rangle_{\infty,CS2I}|}{2|f_{\infty,CS2}|}, \tag{21}$$

$$|\Delta\langle[T_s]\rangle_{\infty,BR2C}| = \chi_{[T_s],\infty,s}|f_{\infty,BR2C,s}| - \chi_{[T_s],\infty,g}|f_{\infty,BR2C,g}|, \tag{22}$$

$$|\Delta\langle[T_s]\rangle_{\infty,BR2C}| = 0. \tag{23}$$
* * *
[9]The precise treatment of the transient proceeds by solving the same inverse problem as outlined in Sec. 2.3, centred around eq. (15), only that the impulse responses in that equation, e.g. $\tilde{h}_{\Psi,g}$, need to be produced as an average from two simulations each, as also done in (Gritsun and Lucarini, 2017).

The subscripts of $\infty$ refer to the asymptotic/stationary climate regime, other subscripts refer to the experiment/forcing scenario. Observe that data from a new experiment is needed, CS2I, where the 'I' indicates an experiment related with CS2 analogously to the relation of SS2I with SS2. Since we are interested in the stationary climate regime only, due to ergodicity we can produce just a single long trajectory instead of an ensemble. The result of this is $\Delta\langle[T_s]\rangle_{\infty,CS2I} = -5.11$ [K] (while we already have $\Delta\langle[T_s]\rangle_{\infty,SS2I} = 4.36$ [K], and from Fig. 2 that $\Delta\langle[T_s]\rangle_{\infty,SS2} = -\Delta\langle[T_s]\rangle_{\infty,CS2} = -4.90$ [K]). Having that $|f_{\infty,BR2C,g}| = |f_{\infty,CS2}|$, we can express the sought-for forcing in relative terms based on the temperature data only, such as:

$$\frac{|f_{\infty,BR2C,s}|}{|f_{\infty,SS2}|} = \frac{|\Delta\langle[T_s]\rangle_{\infty,CS2}| + |\Delta\langle[T_s]\rangle_{\infty,CS2I}|}{|\Delta\langle[T_s]\rangle_{\infty,SS2}| + |\Delta\langle[T_s]\rangle_{\infty,SS2I}|} = 1.08. \tag{24}$$

In fact, we carried out the BR2C experiment independently: *iteratively* determining a solar forcing that cancels to a very good approximation the total response (similarly how the level for e.g. SS2 was determined observing the result of CS2). This forcing in the above relative terms was found to be 1.11, agreeing well with our prediction of 1.08.

Given that our prediction is smaller than the actually needed forcing for cancellation, we can predict an *upper bound* on the actual total response to our predicted forcing by substituting into Eq. (22) the actually needed value $|f_{\infty,BR2C,s}|/|f_{\infty,SS2}| = 1.11$ **(assuming that the response under combined forcing is linear).** This gives $\Delta\langle[T_s]\rangle_{\infty,BR2C} < 0.134$ [K]. Considering that the total residual response with the original methodology (Sec. 2) was 0.6 [K], this means that with the improved methodology we managed to reduce the total response almost to the *one fifth* or even less of the said first result. (Of course, the exact reduction can be easily obtained by an extra simulation, which we have not run.) In fact, some residual total response even with the improved method could be expected, as the simple measure of nonlinearity (17) indicated that linearity is much more 'violated' by increasing radiative forcing as opposed to a reducing one. This prompts that the third-order *odd* perturbative term is not 'minuscule' relative to the second order one – contrary to the assumption of our improved methodology. **Another source of error could be a nonlinear component of the response under combined forcing.**

**4.2 Uncontrolled response and its (non)linearity (II)**

Even if we managed to achieve a perfect cancellation in terms of the global averages, amounting to a success in terms of our objective (I), it is still important to examine the total response in terms of any other observables regarding which the cancellation is not enforced, whether there is any unwanted residual. To this end we look at the BR2C data. In particular, in Fig. 14 we show the spatial variations of the stationary climate in terms of (a) the surface air temperature and (b) annual precipitation. The former one looks like a 'crossover' of Fig. 9 (d) and (f), and the latter like that of Fig. 11 (d) and (f). More precisely, the new diagrams look to lie in between the respective said old diagrams in the sense of an interpolation. This implies that the (true/simulated) variances with respect to space for BR2C ('perfect job'), both for temperature and precipitation, are about the same as those for BR2 ('less than perfect job'), and are much larger than the residual total responses in terms of the respective global averages for BR2. The reason for this is clearly that the response characteristics[10] to greenhouse and solar forcing coinciding with respect to the individual spatial locales are somewhat different. However, it is not really the constancy
* * *
[10]This characteristics is certainly meant to be within the regimes of the actually realised total response. As this regime is finite, possibly significant nonlinear elements of the characteristics are included in our meaning. This is why we did not write at this point 'sensitivity' in place of 'characteristics'.

[Figure]

**Figure 14.** Spatial variation of the stationary climate in terms of (a) the surface air temperature and (b) annual precipitation in the BR2C experiment, when a change in the global average surface air temperature is canceled. (c)**/(d)** The improved linear prediction corresponding to (a)**/(b)**.

of the spatial variance with (slightly) varying levels of the applied solar forcing that is important from a practical point of view, but rather the sensitivity of the response in any locale. Comparing the BR2 and BR2C scenarios, we see that the difference in terms of the climatic surface air temperature could be as much as 2 [K], which is about 10% of the maximal response under the corresponding greenhouse forcing alone.

The improved methodology to estimate susceptibilities applies of course to regional averages too. What remains to be seen now is if linear response theory can predict the residual total responses seen in Fig. 14 (a) and (b) (II). The corresponding linear predictions are shown in panels (c) and (d), respectively. These predictions show a dramatic improvement on the first results shown in Fig. 9 (d) and Fig. 11 (d), respectively. Quantitatively, however, the prediction is not perfect. We can quantify this by e.g. the Pearson correlation coefficient $C$ between the truth $\Delta\langle\Psi\rangle$ and the linear prediction $\langle\Psi\rangle^{(1)}$, the results of which is shown in Table 3. (Note that no weighting of the data points with the area represented by grid points is done.) This shows that the prediction skill is better for the temperature than the precipitation.

Whether the imperfection of the linear prediction is due to nonlinearity – as a small error $E = \Delta\langle\Psi\rangle - \langle\Psi\rangle^{(1)}$ should normally suggest – is not clear, because it is possible that the response $\Delta\langle\Psi\rangle$ is linear but errors in the susceptibility

**Table 3.** Measures of overall nonlinearity of the response in terms of the local temperature and precipitation. $C$ is the Pearson correlation coefficient between the truth $\Delta\langle\Psi\rangle$ and the linear prediction $\langle\Psi\rangle^{(1)}$, and $\rho$ is defined by Eq. (26). Note that to calculate std($\rho$), values of $\rho$ larger in modulus than 5 are discarded. The last column is devoted to the global averages.

|        | Pearson corr. coeff. | std($\rho$) | $\rho$ |
|--------|:--------------------:|:-----------:|:------:|
| $T_s$  | 0.78                 | 0.26        | 0.73   |
| $P_y$  | 0.53                 | 1.01        | 0.70   |

estimates determining $\langle\Psi\rangle^{(1)}$ (or rather its estimator) remain. We should thus find a way to check linearity without relying on the linear prediction. This can be done in a naive way similarly to (17). However, this time we do not have a single forcing present but two. Because of this, it turns out that a check of linearity requires not two but three data points at least. In fact we are readily endowed by three data set candidates resulting from the BR1, BR2 and BR2C experiments. In each scenario, if the response is linear the asymptotic climate would be given by an equation like

$$\Delta\langle\Psi_i\rangle = \chi_{\Psi,g}f_{i,g} + \chi_{\Psi,s}f_{i,s}, \; i = 1,2,3, \tag{25}$$

where $i = 1,2,3$ stand for, say, BR1, BR2, BR2C, in that order. One can express $\chi_{\Psi,s}$ from the eq. of $i = 3$, substitute into the eqs. of $i = 1, 2$, and from these latter express $\chi_{\Psi,g}$. Under linearity the ratio of these expressions,

$$\rho = \frac{\dfrac{\Delta\langle\Psi_2\rangle - \Delta\langle\Psi_3\rangle\frac{f_{2,s}}{f_{3,s}}}{f_{2,g} - f_{3,g}\frac{f_{2,s}}{f_{3,s}}}}{\dfrac{\Delta\langle\Psi_1\rangle - \Delta\langle\Psi_3\rangle\frac{f_{1,s}}{f_{3,s}}}{f_{1,g} - f_{3,g}\frac{f_{1,s}}{f_{3,s}}}}, \tag{26}$$

would be of course unity, meaning that Eqs. (25) are in fact satisfied. We have evaluated $\rho$ for all grid points and display the results in Fig. 15. This suggests that we do have nonlinearity both for the temperature and precipitation. However, this conclusion can be called into question by noticing that the three data points could be too close to one another so that the ratio is not estimated accurately, prompting nonlinearity falsely. One idea to indicate that deviation from unity of both the correlation coefficient $C$ and $\rho$ are due to nonlinearity would be to demonstrate a correlation between the error $E$ of the linear prediction and $\rho$. We have checked the scatter plots of these quantities for both the temperature and precipitation and found no sign of correlations. This, however, does not mean that the response is linear; some unidentified effect can destroy the correlation. Our final idea is that if two situations feature different levels of nonlinearity, even if the two grid-point-wise quantifiers of nonlinearity, $E$ and $\rho$, have random errors, "on average" they should indicate in a coordinated way a stronger deviation from linearity in the case when nonlinearity is actually stronger. We propose to capture this "average" or statistical indicator by the correlation coefficient $C$, on the one hand, and the standard deviation std($\rho$) over the grid points, on the other hand. Clearly, even if linearity is typical, a smaller std($\rho$) would indicate that it is more typical. We have already given the correlation coefficient in Table 3, where we also display std($\rho$). We do indeed see that by both quantities the response of precipitation is prompted to be more nonlinear.

[Figure]

**Figure 15.** Non/linearity of the response in terms of (a) temperature and (b) precipitation, measured by $\rho$ given by the expression (26). Any values of $\rho$ lying outside of the range of the colourbar are represented by the limiting red and blue colours.

In the last column of Table 3 we show $\rho$ for the global averages $[T_s]$, $[P_y]$ (not the average of the grid-point-wise $\rho$'s, but having e.g. $\Psi = [T_s]$ in Eq. (26)). The steady state values are estimated by taking the temporal mean of the ensemble means in the last 80 years. These values could be somewhat inaccurate because of the drift seen in Figs. 4 (b) and 12 (b) for the BR1 simulation. But considering the possible maximum values of $\rho$ for both $\Psi = [T_s]$ and $[P_y]$, a
5    degree of nonlinearity still seems very likely. The figures indicate that the response of the global average precipitation, unlike the local values/regional averages, is not significantly more nonlinear than the response of temperature under geoengineering. These results caution us about the reliability of linear predictions of side effects as part of an assessment exercise;

     – predictions of regional responses are less reliable than the global response, and

10      – some quantities can respond more nonlinearly than others.

**5   Summary and Outlook**

We defined and solved an inverse problem to find a solar forcing that can cancel global warming that would otherwise result from a change in the greenhouse forcing. In fact, we can allow for other choices of the scalar observable **to keep under control**, either with respect to the physical quantity, or considering e.g. local variables. **One can also prescribe an arbitrary time**
15   **evolution of the chosen observable. The inverse problem constitutes thereby a generic framework for analysing/assessing geoengineering scenarios.** The inverse problem itself was derived in the framework of linear response theory. Because of the true nonlinear characteristics of the response the degree of approximation of the solution specifically for the cancellation of global average surface air temperature depended on the method and its success of determining the linear susceptibilities or Green's functions belonging to the different forcings (I). The issue stems from the fact that for the estimation of the Green's
20   functions we used *finite* magnitude external system identification forcings, in which case the nonlinearity of the response is already felt, while for the cancellation, i.e., *zero* total response, we would need the linear susceptibilities *exactly* **– assuming**

**the response is linear under combined forcing**. An inaccurately predicted required solar forcing leads to a nonzero residual true total response.

By a simple method, also used in (Gritsun and Lucarini, 2017), here, for determining the susceptibilities, we eliminate even-order nonlinearities from the response in the system identification experiments. The price of this is having to run double as many simulations for system identification. In the scenario of doubling $CO_2$ concentration, by this method we could cut five-fold the unwanted actual total response arising instead of cancellation. Furthermore, the linear prediction of spatial patterns using the improved *local* susceptibilities improved dramatically. **Nevertheless, the prediction is not perfect, and we indicated that the response under combined forcing should be somewhat nonlinear, and the degree of nonlinearity could be typically stronger for some quantities. In particular, we found that in PlaSim the response of precipitation is more nonlinear than that of the surface temperature. This casts a shadow over the use of response theory for an efficient assessment. Perhaps there would be still value in this method as larger scale quantities are expected to be better predictable. It may also be that the nonlinearity is more modest in complex models. Otherwise it would be desirable in the future to work out a method of predicting the nonlinear response in geoengineering scenarios.**

**Ours is the first such analysis of the linearity of regional response under geoengineering. It is a question whether our findings in PlaSim carry over to state-of-the-art Earth System Models because they do respond more weakly in the presence of the seasonal cycle. The question certainly seems valid, however, as also CMIP5 models do feature nonlinear regional response under [$CO_2$] forcing only (Good et al., 2015; Winton, 2013). The response of global average surface air temperature and precipitation has been found by MacMartin and Kravitz (2016) approximately linear in some CIMP5 models, seemingly more so than in PlaSim, but weaker forcing than [$CO_2$]-doubling was considered, and the linearity of regional responses were not analysed in detail.**

We pointed out also that instead of step-wise system identification forcing, it is better to use a Kronecker delta forcing in order to achieve a better signal-to-noise ratio. As another gain from using a Kornecker delta forcing, the response would be much more modest in magnitude, and hence it would stay further off regimes with more significant contributions of nonlinear terms, and so the linear susceptibilites could be estimated more accurately even by the naive method.

We note that the presented method of predicting a required solar forcing is based on Green's functions that are determined by externally forcing the system of interest. This is clearly not a method that could be put in practice in the case of the Earth system. Therefore, this is another reason, beside the unpredictability of the 21st century greenhouse forcing, why the method is suitable only for scenario analyses. However, the Green's functions might be possible to estimate without externally forcing the system, just from an observation of unforced fluctuations. The crucial question in this regard is whether the fluctuation-dissipation theorem (Kubo, 1966; Leith, 1975) is applicable.

**Appendix: The circular convolution theorem and its application**

Taking the discrete-time Fourier transform (DTFT) of Eq. (12) we have, via the convolution theorem for discrete sequences (Katznelson, 1976), a formally analogous version of Eq. (6) with the individual Fourier transforms approximated by Fourier series:

$$\langle\hat{\Psi}\rangle_{2\pi}^{(1)}(\omega) = \hat{\chi}_{\Psi,2\pi}(\omega)f_{2\pi}(\omega), \tag{1}$$

where e.g. $f_{2\pi}(\omega) = \mathrm{DTFT}\{Tf[n]\} = \sum_{n=-\infty}^{\infty} Tf[n]e^{-i\omega n}$ and $f[n] = \mathrm{DTFT}^{-1}\{T^{-1}f_{2\pi}(\omega)\} = \frac{1}{2\pi T}\int_{-\pi}^{\pi} d\omega f_{2\pi}(\omega)e^{i\omega n}$ with a normalised nondimensional angular frequency $\omega$. Featuring instead the dimensional frequency $f$ measured in Hertz = $[\sec^{-1}]$, the forward and inverse transformation pairs are symmetrical: $f_{1/T}(f) = f_{2\pi}(2\pi fT) = \sum_{n=-\infty}^{\infty} Tf[n]e^{-i2\pi fTn}$ and $f[n] = T\int_{1/T} df f_{1/T}(f)e^{i2\pi fTn}$. The DTFT, a continuous function of the frequency $f$, is often sampled at $f = k/(NT)$, $k = 0,\ldots,N-1$:

$$f_{1/T}(k/(NT)) = T\sum_{n=-\infty}^{\infty} f[n]e^{-i2\pi kn/N} = T\sum_{n=n_0}^{n_0+N} f_N[n]e^{-i2\pi kn/N} = T\times\mathrm{DFT}\{f_N[n = n_0,\ldots,n_0+N]\}, \tag{2}$$

with any $n_0$, which yields the discrete Fourier transform (DFT) of the *finite* sequence $f_N[n]$, $n = n_0,\ldots,n_0+N$, where the full infinite sequence $f_N[n]$, $n \in \mathbb{R}$, turns out to be $N$-periodic, since for the equivalence of the two sums under (2) it has to be in the so-called periodic summation form:

$$f_N[n] = \sum_{m=-\infty}^{\infty} f[n-mN]. \tag{3}$$

Therefore, when $f[n]$ is actually $N$-periodic, its DTFT is nonzero only at $f = k/(NT)$, $k \in \mathbb{R}$, and also periodic, such that the DFT of a single cycle of $f[n]$ is able to represent its DTFT. For such periodic sequences, to be denoted distinctively using a subscript as $f_N[n]$, it can be proven (Katznelson, 1976) that:

$$y * f_N = \mathrm{DTFT}^{-1}\{\mathrm{DTFT}\{y\}\mathrm{DTFT}\{f_N\}\} = \mathrm{DFT}^{-1}\{\mathrm{DFT}\{y_N\}\mathrm{DFT}\{f_N\}\}, \tag{4}$$

with any nonperiodic sequence $y[n]$. Note that $y * f_N$ is referred to as the *circular convolution* of sequences $y[n]$ and $f[n]$. When the $y[n]$ and $f[n]$ sequences have a finite length, $n = 0,\ldots,N-1$ with any $N \geq 1$, so that e.g. $f_N[n] = f[\mathrm{mod}(n,N)]$, their circular convolution can be shown (Katznelson, 1976) (https://uk.mathworks.com/help/signal/ug/linear-and-circular-convolution. html) to be:

$$(y * f_N)[n = 0,\ldots,N-1] = \sum_{k=0}^{N-1} y[k]f_N[n-k] = \mathrm{DFT}^{-1}\{\mathrm{DFT}\{y\}\mathrm{DFT}\{f\}\}, \tag{5}$$

which equality is called the *circular convolution theorem*. It follows that when $y[n] = 0$ and $f[n] = 0$ for $n = 0,\ldots,N_f - 1$ and $N_y - 1$, respectively, then $(y * f_N)[\mathrm{mod}(n-1,N)] = (y * f)[n]$ for $n = N,\ldots,N + \min(N_f + N_y, N-1)$. Furthermore, $(y * f)[n]$, $n = 1 + N_f + N_y,\ldots,N + 1 + N_y$ is the segment that represents the part of the linear convolution that can be considered useful in the sense that it coincides with the occurrence of the finite values of $f$ in a finite time interval of length

$N - N_f$. Therefore, the circular convolution $(y * f_N)[n]$ captures the useful part of the linear convolution over $n = \max(1 + N_f + N_y, N), \ldots, N + \min(1 + N_y, N_f + N_y, N - 1)$.

Therefore, when facing the practical situation of having *finite* time series, $f[l]$ and $h_\Psi[l]$, $l = 0, \ldots, L-1$, Eq. (5) can be used to determine the response $h_\Psi * f[l]$, $l = 0, \ldots, L-1$ (whose usefulness is coming from efficient algorithms for evaluating the DFT, called fast Fourier transform algorithm, FFT). In particular, if the two sequences are to be *padded* in front by a number $N_f = N_h = N_0$ of zeros equally (so that the circular convolution (5) be well-defined), then the reconstructed length of the linear convolution $h_\Psi * f$ (the response of a causal system coinciding with the forcing) is $1 + N_0 - \max(N_0 - L + 1, 0)$. This is a linear function of $N_0$ saturating at $N_0 = L-1$ reaching the full length $L$. Therefore, for simplicity one can pad by $N_0 = L-1$ zeros[11], and we will *denote these padded sequences* by e.g. $\tilde{f}[l]$, $l = 0, \ldots, 2(L - 1)$. Note that padding with fewer or no zeros results in a circular convolution that better approximates either the useful or the not useful part of the linear convolution, which approximation is the better the more zeros are used. In the extreme case of no padding, very little of the useful part could be well-approximated. The key to the applicability of Eq. (4) is that it does not matter how the forcing $f[n]$ – and with it the response $\langle \hat{\Psi} \rangle [n]$ – continue after our experiment, and so they can be thought of as periodic.

*Competing interests.*  The authors declare to have no competing interests.

*Acknowledgements.*  The authors would like to express their gratitude to an anonymous referee for their very thorough feedback and many suggestions to improve the quality, and also for providing us references of the relevant literature. This work is part of the EU Horizon 2020 project CRESCENDO (under grant No. 641816); the financial support is gratefully acknowledged. It also received support from the EU Blue Action project (under grant No. 727852); and VL acknowledges support from the DFG Sfb/Transregio TRR181 project.
* * *
[11] This results in an odd sequence length, which has an adverse effect on the common fft algorithm performance. Therefore, in actual practice one can produce time series data of length $L$ being some power of 2, and pad by an equal number of zeros.

---

## Referee Comment (RC2) · Anonymous Referee #2 · 26 Aug 2018

I think this paper contains the germ of a good idea. It would be interesting to plot an actual, rather than schematic version of Figure 1, for a large number of interesting observables (beyond global mean temperature and precipitation to include some integral measures of, for example, economic or ecosystem damage). However, the writing is so difficult to read, the emphasis on what turns out to be a not-very-useful linear inverse technique for estimating what solar forcing trajectory would be required to cancel out a given greenhouse gas forcing trajectory is so over-done, and the absence of any figure like figure 1 using actual data, that I don't think it is publishable in this form.

The key problem with the paper as presented is that finding the solar forcing trajectory that would cancel an exponential increase of $CO_2$ seems to be quite easy in the simplified climate model the authors use: just iterate the value of the solar constant to

zero out the global mean temperature rise at a large value of CO2, and then adjust the solar forcing linearly towards that solution over the ramp period. Since numerous perturbation experiments need to be performed to solve the inverse problem, it's not clear what benefit that more mathematically complicated procedure provides. Furthermore, the real problem with geoengineering is that you can't exactly cancel the CO2-forced climate change at all points using a single means of controlling the solar cycle, and this paper provides no help in achieving that more thorough cancellation, nor in documenting the already well-known difficulty.

The first reviewer has covered many of the points I would have made (and several more that I didn't think of, not knowing the geoengineering literature as well), but I will add a few specific points: 1) the concept of system identification is used many times, but was never defined in a comprehensible way. 2) "primarily of a diplomatic nature" is a distracting comment, not clear what's intended and why its important in this context. 3) The section on response theory is especially murky. It's not clear enough (e.g. all symbols in equations are not defined as soon as they're introduced) to allow a reader unfamiliar with the math to actually learn it, and it seems to be based on already published material, so it would be better to just refer interested readers to more thorough descriptions published elsewhere. The phrase "of course" is used frequently through the paper when derivations are being skipped and the conclusions being described are not at all obvious. I would also suggest banning the word "obvious". 4) Page 13, L12, not clear what "aymptotic times" refers to.

---

## Short Comment (SC1) · 26 Aug 2018

In going through the citation list for this paper, I noticed that a paper that focused on linearity of climate response in the geoengineering context was not cited. I know that it is not the best form to ask people to consider your own papers, but in this case I think the suggestion may be warranted.

Several papers that Long Cao has led have discussed issues associated with the linearity of the climate response to solar geoengineering, but from a more empirical and less mathematical perspective than the work under consideration, and also not in the context of control theory. Nevertheless Bódai et al may want to consider the

following paper:

https://agupubs.onlinelibrary.wiley.com/doi/abs/10.1002/2015JD023901

Fast and slow climate responses to CO2 and solar forcing: A linear multivariate regression model characterizing transient climate change
Long Cao, Govindasamy Bala, Meidi Zheng, and Ken Caldeira
Geophysical Research Letters
20 November 2015 https://doi.org/10.1002/2015JD023901

Apologies in advance for burdening the authors with this shameless self-promotion.

―――――――――――

---

## Short Comment (SC2) · 26 Aug 2018

Sorry, citation in the previous comment should have read:

https://agupubs.onlinelibrary.wiley.com/doi/abs/10.1002/2015JD023901

Fast and slow climate responses to CO2 and solar forcing:  A linear multivariate regression model characterizing transient climate change
Long Cao, Govindasamy Bala, Meidi Zheng, and Ken Caldeira
JGR-Atmospheres 20 November 2015
https://doi.org/10.1002/2015JD023901

---

## Author Comment (AC2) · 20 Sep 2018

We are surprised by the referee's comments and verdict. Those comments that form the basis of the referee's recommendation against publication seems to overlook content of the manuscript and key parts of the discussion between referee #1 and ourselves.

It is suggested that our paper is not suitable for publication for two reasons. Both of them can be discounted as follows.

1. Instead of a schematic for Fig. 1 we are required to present a diagram representing "actual data". It is unclear what is meant by "actual data". Surely it cannot be observational data because no geoengineering is practiced presently and in the recent past. In

[Figure]

the context of our model PlaSim, Figs. 4, 7, 8, 9, 10, 11, 12, 14 all present side-effects of geoengineering — what Fig. 1 is meant to give a first idea of.

2. The solution for the required solar forcing of the inverse problem, given a ramp CO2 forcing, is rather trivial, so the methodology that we are proposing (point (I) of the abstract) is unnecessary. The referee does not realise that — as we emphasized this in Sec. 1.2 — cancelling the global mean temperature is just one of the possible choices for a combination of an observable to control & a desired outcome. For other choices, which are not necessarily unrealistic, the solution could be nontrivial. Examples may be the average temperature where people live, or, where the country of an influential nation is located.

Regarding point 2. above, the referee outlines the way the required solar forcing can be determined for our ramp CO2 forcing, involving an iterative determination of the constant-in-time solar forcing of the plateau, after the ramp. This is exactly what we wrote staring on line 15 on page 24 of the original submission.

The following statement from the referee report likely pertains to the same issue.

"the real problem with geoengineering is that you can't exactly cancel the CO2-forced climate change at all points using a single means of controlling the solar cycle, and this paper provides no help in achieving that more thorough cancellation, nor in documenting the already well-known difficulty."

The said "problem" is what our original submission meant to exposit (starting with Fig. 1!), and referee #1 pointed out to be the most obvious thing about geoengineering and already thoroughly addressed. This has been acknowledged by us, and the paper had been revised.

We find the following four points of the referee also surprising, and we respond to them as follows. 1) Using the term "identification forcing" systematically, and explaining how via such a forcing the Green's function is determined, having already explained what

the Green's function is needed for, we are satisfied that our intended message can be understood by a devoted reader. A "comprehensive" description of "system identification" appears to us a subjective concept, and, again, it's not clear to us how our message or exposition would be strengthened by further details on system identification in general. 2) We wish not to remove this comment. Regarding the clarity of our meaning, we had removed a footnote in the revision, as a result of the criticism by referee #1. In that we distinguished between societal challenges of diplomatic and technological nature. 3) The referee will find many papers in prestigious journals which adopt as much detail from published papers as we did. He/she will find examples among the references cited by us. The liking of expressions "of course" and "obvious" is subjective. In our case we think that it does not obstruct the understanding and neither are they misleading. We assume when using these expressions a degree of dedication of the reader. 4) Although the expression "asymptotic time" is not defined, we think that the second part of the sentence "the discrepancy emerges transiently only" makes the meaning clear enough, provided that the reader know the meaning of "transient".